# Short antisense oligonucleotides alleviate the pleiotropic toxicity of RNA harboring expanded CGG repeats

Magdalena Derbis[1], Emre Kul [2], Daria Niewiadomska [1], Michał Sekrecki[1], Agnieszka Piasecka[1], Katarzyna Taylor[1], Renate K. Hukema [3,4], Oliver Stork[2] & Krzysztof Sobczak[1✉]

Fragile X-associated tremor/ataxia syndrome (FXTAS) is an incurable neurodegenerative disorder caused by expansion of CGG repeats in the *FMR1* 5'UTR. The RNA containing expanded CGG repeats (rCGG$^{exp}$) causes cell damage by interaction with complementary DNA, forming R-loop structures, sequestration of nuclear proteins involved in RNA metabolism and initiation of translation of polyglycine-containing protein (FMRpolyG), which forms nuclear insoluble inclusions. Here we show the therapeutic potential of short antisense oligonucleotide steric blockers (ASOs) targeting directly the rCGG$^{exp}$. In nuclei of FXTAS cells ASOs affect R-loop formation and correct miRNA biogenesis and alternative splicing, indicating that nuclear proteins are released from toxic sequestration. In cytoplasm, ASOs significantly decrease the biosynthesis and accumulation of FMRpolyG. Delivery of ASO into a brain of FXTAS mouse model reduces formation of inclusions, improves motor behavior and corrects gene expression profile with marginal signs of toxicity after a few weeks from a treatment.

_______________

[1] Department of Gene Expression, Institute of Molecular Biology and Biotechnology, Adam Mickiewicz University, Uniwersytetu Poznanskiego 6, Poznan, Poland. [2] Department of Genetics and Molecular Neurobiology, Institute of Biology, Otto von Guericke University, Magdeburg, Germany. [3] Department of Clinical Genetics, Erasmus MC, CA Rotterdam, The Netherlands. [4]Present address: Department of Health Care Studies, Rotterdam University of Applied Sciences, HR Rotterdam, The Netherlands. ✉email: ksobczak@amu.edu.pl

Several genetic disorders caused by the expansion of trinucleotide CGG repeats (CGG[exp]) in the 5′ untranslated region (5′UTR) of the *FMR1* gene located on chromosome X have been described. The most severe is fragile X syndrome (FXS), which is the most common cause of intellectual disability and autism in men. FXS results from large expansions containing over 200 CGG repeats, which cause complete silencing of the *FMR1* gene and, therefore, a lack of fragile X mental retardation protein (FMRP), which plays an important role in synaptic plasticity[1]. Milder expansions with 55-200 CGG repeats lead to the development of either fragile X-associated primary ovarian insufficiency (FXPOI), which manifests as premature menopause, or a late-onset neurodegenerative disorder called fragile X-associated tremor/ataxia syndrome (FXTAS)[2–4]. Due to incomplete penetrance of the mutation, approximately 1 in 5000 men older than 50 suffer from FXTAS[4]. Random inactivation of chromosome X in women greatly reduces the risk of disease development[4]. The main clinical features of FXTAS include intention tremor, gait ataxia, and cognitive decline. Neuropathy, mild brain atrophy, and white matter lesions develop progressively in the presence of ubiquitin-positive intranuclear inclusions in neurons and astrocytes accompanied by a twofold to eightfold increase in the *FMR1* mRNA level[2–5].

The molecular basis of FXTAS is still incompletely understood, but three main pathomechanisms of this disorder have been proposed (Fig. 1a). First, as the 5′UTR region of *FMR1* is GC-rich, the cotranscriptional formation of R-loop structures can be induced[6–8]. These RNA/DNA hybrids formed by newly synthesized RNAs containing expanded CGG repeats (rCGG[exp]) and complementary DNA strands containing CCG repeats may induce local DNA damage causing cell death[3,6,8]. R-loop formation depends on the CGG repeat length, hence, CGG[exp] hairpins stabilize this structure substantially[8]. Second, mutated *FMR1* mRNA can interact with many nuclear RNA-binding proteins (RBPs) and form inclusions in the nucleus of neurons in the brain of patients with FXTAS [9–16]. This mechanism is known as RNA toxicity. The sequestration of proteins such as DGCR8 (DiGeorge Syndrome Critical Region 8) and SAM68 (Src-associated in mitosis 68 KDa protein) interferes with miRNA maturation and the alternative splicing of SAM68-dependent genes, respectively[13,15]. Third, repeat-associated non-AUG (RAN) translation of rCGG[exp] can yield toxic polyglycine-containing proteins (FMRpolyGs) that are prone to form nuclear inclusions in FXTAS patients' brains[17–19]. The sequence of the 5′UTR of *FMR1* carrying rCGG[exp] is translated beginning at near-cognate ACG or GUG start codons upstream of the CGG repeats. In addition, the natural product of this gene, FMRP, is produced starting at a canonical start codon downstream of the CGG repeats within a different reading frame[18,19].

The three proposed pathomechanisms have one molecule in common, rCGG[exp], suggesting that FXTAS could be highly amenable to RNA targeting interventions; however, no curative therapy is currently available. A few reports have suggested that small compounds targeting CGG[exp] within *FMR1* mRNA have therapeutic potential. In an FXTAS cell model, some RNA-binding compounds reduced SAM68-dependent aberrant splicing and inhibited RAN translation of FMRpolyG[20–22]. In a similar cell model, we showed that antisense oligonucleotides (ASOs) complementary to the *FMR1*-specific sequence downstream of the CGG repeat tract induced targeted mRNA degradation through RNase H1 recruitment and caused a significant decrease in FMRpolyG production and aggregation[23]. Recently, ASOs targeting RAN translation initiation site on mRNA was also proved to reduce biosynthesis of FMRpolyG in FXTAS cellular models[24]. However, none of these compounds have been tested in vivo.

The aim of this work was to determine the therapeutic potential of short chemically modified ASOs directly targeting rCGG[exp] in *FMR1* mRNA. We used ASO steric blockers that bind to complementary sequences without inducing mRNA degradation and reduce the overall toxic effect of expanded repeats. Here, we present evidence that short ASOs complementary to rCGG[exp] (ASO–CCG) can alleviate not only molecular hallmarks of FXTAS, such as cotranscriptional R-loop formation, nuclear rCGG[exp]-mediated toxicity, and FMRpolyG biosynthesis but also FXTAS-related molecular and behavioral phenotypes after long-lasting infusion of ASO–CCG directly into a brain ventricle in a mouse model of FXTAS.

## Results

### ASO–CCG binds rCGG[exp] with high affinity and releases nuclear proteins from sequestration.

Pure CGG repeat tracts in RNA (rCGGs) form the most stable hairpin structure of all trinucleotide RNA repeats (e.g., rCUG, rCAG, and rCCG), with a melting temperature of ~80 °C[25]. Moreover, rCGG hairpins are further stabilized by flanking sequences within *FMR1* mRNA[26] and are thus not readily available to interact with other nucleic acids. Therefore, ASOs designed to bind to rCGG repeats should be characterized by a higher affinity and thermodynamic stability for ASO/RNA duplexes formed in trans than for RNA/RNA hairpins formed in cis. Thus, we designed ASO blockers with locked nucleic acid-modified phosphorothioate oligonucleotides (LNA-PSs) and CCG repeat sequences (i.e., ASO–CCG). One oligomer has 9 LNA residues; the other, 11. As a control, we used ASO-ctrl with a random sequence, with the same chemical modifications and corresponding lengths (either 9 or 11 nt).

As a first step, we sought to determine whether different types of ASO–CCG can efficiently invade the rCGG hairpin in vitro. We performed an electrophoretic mobility shift assay (EMSA) in which different concentrations of two ASO–CCGs were used to calculate their affinity for radioactively labeled rCGG$_{20}$. Both ASO–CCGs exhibited high affinity, especially the shorter ASO variant, with a dissociation constant ($K_d$) in the high picomolar range (Fig. 1b and Supplementary Fig. S1a). Next, we sought to determine whether ASO–CCGs can efficiently enter living cells. Based on our previous study[27] testing ASOs of different sequences but composed exclusively of LNA units, we anticipated that ~20–30% of ASO–CCG may enter treated cells. Fluorescently labeled ASO–CCGs delivered via lipofection passed through the cell membrane, producing a unique dispersed signal in cells (Supplementary Fig. S1b). Moreover, the results of fluorescence microscopy suggested that almost all cells received comparable amounts of ASO–CCG. We checked also the general toxicity of tested ASOs in comparison to an ASO showing significant sequence-related toxicity used as a positive control (toxic ASO). During the time-course experiment, the viability of cells treated with ASO–CCG and ASO-ctrl was similar to mock-treated cells (Fig. 1c). In addition, a flow cytometry assay revealed that short LNA-based ASOs did not contribute to apoptosis (Fig. 1d). Therefore, we assumed that both control and rCGG repeat-targeting oligonucleotides analyzed in this study show no or very low cell toxicity.

The high affinity and bioavailability, as well as low cellular toxicity of ASO–CCGs, encouraged us to examine their effects on (1) RNA toxicity, (2) RAN translation efficiency, and (3) cotranscriptional R-loop formation in different FXTAS cell models. First, we tested whether ASO–CCGs treatment could rescue the activity of the nuclear proteins DGCR8 and SAM68, involved in miRNA processing and RNA splicing, respectively, which are sequestered by the hairpin structures formed by rCGG[exp]. We used artificial FXTAS cell models in which CGG

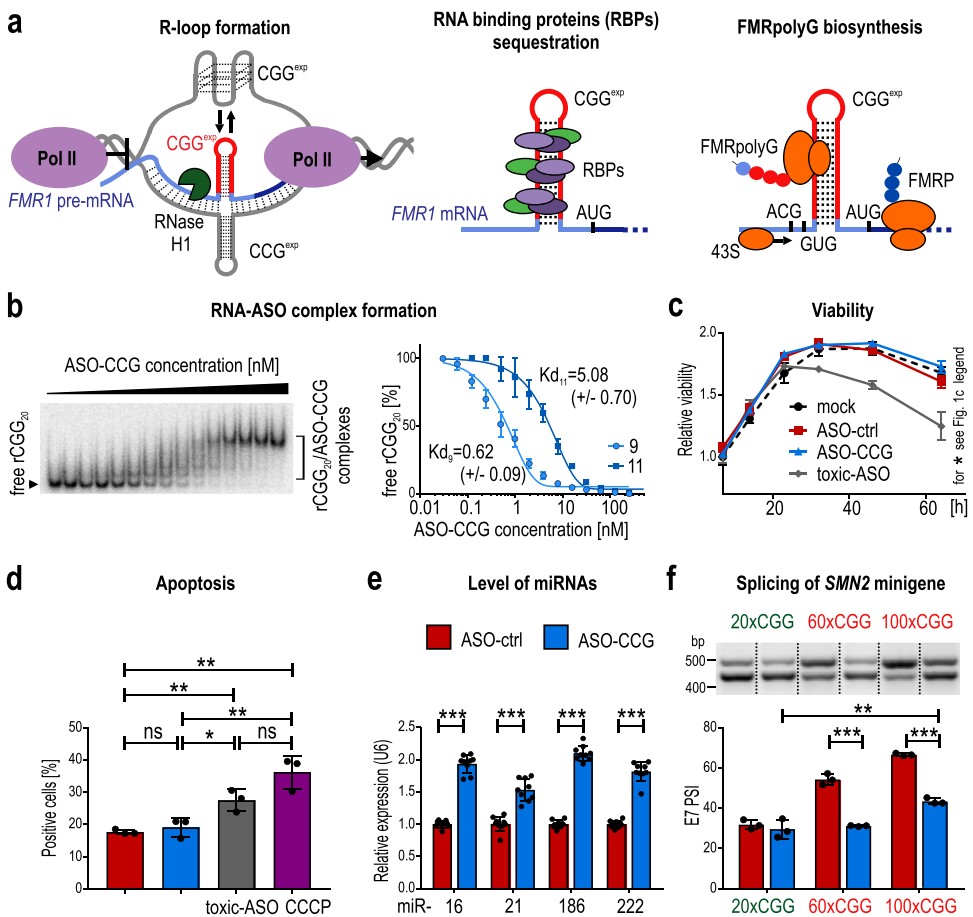

**Fig. 1 Binding of ASO–CCG to CGG^exp in vitro and in cells reduces the effects of toxic RNA. a** Potential molecular pathomechanisms of FXTAS. **b** EMSA of the interaction of rCGG$_{20}$ with ASO–CCGs having 9 and 11 LNA residues. The graph presents means with SDs. $N = 3$ independent samples for each concentration. Kd, a dissociation constant [nM]. Light blue/dots, 9 nt; dark blue/squares, 11 nt ASO. **c** Influence of ASOs (9 nt, 200 nM) on COS7 cells viability. Graph presents mean of $N = 4$ biologically independent samples with the SDs. Black/dots, mock (transfection agent); red/squares, ASO-ctrl; blue/tringles, ASO–CCG; dark gray/diamonds, toxic ASO. For 64-h time point: mock vs ASO-ctrl, $P = 0.069$; mock vs ASO–CCG, $P = 0.187$; mock vs toxic ASO, $P = 0.0004$; ASO-ctrl vs ASO–CCG, $P = 0.012$; ASO-ctrl vs toxic ASO, $P = 0.001$; ASO–CCG vs toxic ASO, $P = 0.0002$. **d** Influence of ASOs (9 nt, 200 nM) on apoptosis of COS7 cells. Graph presents mean of $N = 3$ biologically independent samples with the SDs. Dark gray bar, toxic ASO; purple bar, carbonyl cyanide 3-chlorophenylhydrazone (CCCP, positive control). **e** RT-qPCR quantification of the level of endogenous miRNAs in COS7 cells overexpressing rCGG^exp and treated with ASOs (9 nt, 200 nM). Graph presents mean of $N = 3$ biologically independent samples (each with $n = 3$ technical replicates) with the SDs. **f** RT-PCR analysis of SAM68-dependent alternative splicing of exon 7 in *SMN2* minigene in COS7 cells overexpressing normal (green) or expanded CGG repeats (red) and treated with ASOs (11 nt, 200 nM). Graph presents mean of $N = 3$ biologically independent samples with the SDs. 100×CGG ASO-ctrl vs 100×CGG ASO–CCG, $P = 0.00002$. The upper bands, the exon 7 inclusion isoforms; the lower bands, the exon 7 exclusion isoforms. PSI, the percent of spliced in. **b, f** Samples were derived from the same experiment and processed in parallel on different gels. Presented gels were cropped. **d–f** Red bars, ASO-ctrl; blue bars, ASO–CCG. **c–f** Two-sided, unpaired Student's *t* test; *$P < 0.05$; **$P < 0.01$; ***$P < 0.001$; ns, non-significant. **b–f** Source data are provided as a Source Data file.

repeats with normal (~20×CGG) and pathogenic (~60×CGG and ~100×CGG) lengths were overexpressed[13,15]. Overexpression of RNA containing CGG^exp but not normal-length CGG repeats caused significant downregulation of several endogenous mature miRNAs, indicating that DGCR8 was likely sequestered in the latter cells (Supplementary Fig. S1c, d). Furthermore, treatment with ASO–CCG but not with ASO-ctrl significantly increased the levels of mature miRNAs, suggesting that DGCR8 activity was restored in treated cells (Fig. 1e). Similar experiments were performed to assess SAM68 splicing factor activity, by measuring alternative splicing of the exon 7 of *SMN2*. Cells expressing expanded repeats of ~60 or ~100 CGGs but not ~20 CGGs showed increased exon 7 inclusion in the *SMN2* minigene mRNA (Fig. 1f and Supplementary Fig. S1e, f). Treatment with ASO–CCG but not ASO-ctrl abolished this effect, leading to partial (100×CGG) or even full (60×CGG) splicing correction

(Fig. 1f and Supplementary Fig. S1f). By contrast, ASO–CCG treatment did not affect the alternative splicing of SAM68-independent event, which proved the specificity of the observed effect (Supplementary Fig. S1g). Overall, these results indicate that the interaction of ASO–CCG with rCGG^exp in cell nuclei can release DGCR8 and SAM68 from pathogenic sequestration, leading to a significant reduction in the effects of toxic RNA in the FXTAS cell model.

**ASO–CCG reduces the efficiency of FMRpolyG biosynthesis.** Next, we investigated whether ASO–CCG can reduce the level of toxic FMRpolyG. We overexpressed a 100×CGG construct carrying the 5′UTR sequence of the *FMR1* gene containing ~100 CGG repeats (Fig. 2a; artificial FXTAS model)[19,23]. RAN translation of mRNA from this construct leads to the biosynthesis of FMRpolyG fused with GFP (FMRpolyG-GFP). Since FMRpolyG-

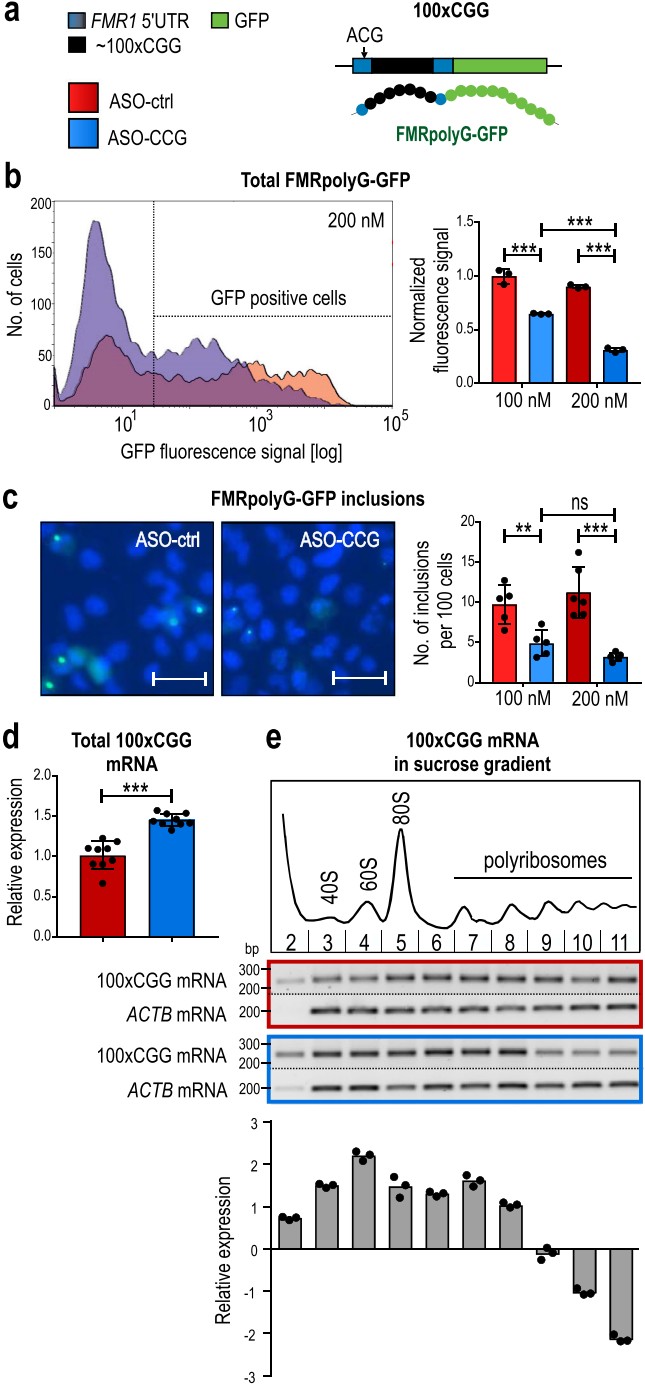

**Fig. 2 The effect of ASO-CCG on FMRpolyG biosynthesis and aggregation. a** Schematic of the 100×CGG genetic construct. This construct contains the 5′UTR of the *FMR1* gene (blue bars) with ~100 CGG repeats (black bar) fused with the GFP coding sequence (green bar). Biosynthesis of the FMRpolyG-GFP via RAN translation starts from the near-cognate start codon, ACG. **b** Cytometric quantification of the total level of FMRpolyG-GFP in COS7 cells expressing 100xCGG construct and treated with ASOs (11 nt). The fluorescence signal of GFP-positive cells was measured, excluding dead cells stained with propidium iodide. The histogram presents different FMRpolyG-GFP signal distribution in cells treated with ASO-ctrl (red) or ASO-CCG (blue). $N = 3$ biologically independent samples. ASO-ctrl 200 nM vs ASO-CCG 200 nM, $P = 0.000003$. **c** Microscopic quantification of FMRpolyG-GFP inclusions in COS7 cells expressing 100×CGG construct and treated with ASOs (11 nt). Representative images were pseudo-colored and merged; green, GFP-positive inclusions; blue, nuclei stained with Hoechst 33342; scale bars, 50 μm. $N = 6$ biologically independent samples for ASO-ctrl 200 nM and $N = 5$ for other conditions. ASO-ctrl 200 nM vs ASO-CCG 200 nM, $P = 0.0003$. **d** RT-qPCR quantification of total mRNA from 100xCGG construct in COS7 cells treated with ASOs (9 nt, 200 nM). $N = 3$ biologically independent samples (each with $n = 3$ technical replicates). **e** Association of 100xCGG mRNA with polyribosomes. Extracts from COS7 cells transfected with the 100×CGG construct and 200 nM ASOs (9 nt) were fractionated on a linear sucrose gradient (15–45%). The total RNA was isolated from the collected fractions representing free mRNAs (the first fraction), monoribosomes, and polyribosomes. Cropped gel presents RT-PCR results. The graph presents RT-qPCR results with a mean of $n = 3$ technical replicates with the SDs. The reference sample is the corresponding fraction in ASO-ctrl-treated cells and the reference gene is *GAPDH*. The experiment was repeated 2 times with similar results. **b–d** Red bars, ASO-ctrl; blue bars, ASO-CCG; light colors, 100 nM; dark colors, 200 nM. Graphs present means of indicated $N$ with the SDs. Two-sided, unpaired Student's $t$ test; $*P < 0.05$; $**P < 0.01$; $***P < 0.001$; ns, non-significant. **b–e** Source data are provided as a Source Data file.

separation of cell lysates. ASO-CCG significantly reduced the level of both the soluble FMRpolyG-GFP and FMRP equivalent (Supplementary Fig. S2c).

We then investigated whether ASO-CCG reduced FMRpolyG-GFP levels by inducing mRNA degradation or by reducing the mRNA translation rate. Measurement of the total level of rCGG$^{exp}$ showed significantly higher levels in ASO-CCG-treated cells compared to cells treated with ASO-ctrl ruling out an increase in mRNA degradation (Fig. 2d). Next, to evaluate the dynamics of mRNA translation, we performed sucrose-gradient fractionation of cell lysates after treatment with ASO-CCG. Compared to ASO-ctrl, ASO-CCG treatment increased the abundance of mRNA containing CGG$^{exp}$ in the fractions corresponding to free mRNAs, monoribosomes, and light polyribosomes and decreased the association of this mRNA with actively translating heavy polyribosomes (Fig. 2e). This finding suggests that despite the increase in the steady-state level of mRNA containing CGG$^{exp}$, ASO-CCG interfered with its translation process by binding to rCGG$^{exp}$ and preventing its association with polyribosomes, leading to a significant reduction in the FMRpolyG-GFP level.

Short CGG repeat tracts with at least six uninterrupted repeats are present in ~1.5% of human mRNAs. These repeat tracts may contain as many as 16 pure CGG repeats, and most (~60%) are located in 5′UTRs[28]. Therefore, we investigated whether ASO-CCG can influence the level and polyribosome association of transcripts containing short CGG repeats. We evaluated endogenous *FMR1* and *NUB1* mRNAs, which are predicted to contain 12 CGGs repeats. The total level of both transcripts was

GFP is present in cells in both a soluble and an aggregated form, we used three different methods, based on fluorescence signal quantification, to measure total, aggregated, and soluble forms of this protein[23]. Using a cytometric assay, we showed that the total amount of FMRpolyG-GFP produced in cells expressing 100×CGG was significantly reduced in cells treated with ASO-CCG compared to that found in the same cells treated with ASO-ctrl, and that this effect was concentration-dependent (Fig. 2b and Supplementary Fig. S2a, b). Using fluorescent microscopy, we found that ASO-CCG treatment resulted in a significant reduction in the number of inclusions of aggregated protein compared to cells treated with ASO-ctrl (~3 and ~11 per 100 nuclei, respectively; Fig. 2c). We also quantified the newly synthesized soluble form of FMRpolyG-GFP as well as FMRP equivalent using fluorescence signal detection after SDS-PAGE

elevated in ASO–CCG-treated cells (Supplementary Fig. S2d) and their distribution in the monoribosome/polyribosome fractions was changed compared to ASO-ctrl-treated cells, with a decreased abundance on heavy polyribosomes (Supplementary Fig. S2e). These results suggest that ASO–CCG influences the total level of transcripts containing not only expanded but also short CGG repeats and, possibly through binding to the 5′-end of these mRNAs, may decrease their translation rate.

**ASO–CCG increases the transcription of the *FMR1* locus and reduces the efficiency of FMRP biosynthesis.** ASO–CCG increased the level of exogenous mRNAs containing $CGG^{exp}$ and transcribed from the genetic construct in the artificial FXTAS model. Therefore, we asked whether ASO–CCG exerts a similar effect on endogenous *FMR1* mRNA and pre-mRNA in human fibroblasts. Moreover, it had been previously shown that R-loops in 5′-parts of other genes may reduce transcription efficiency[29]; therefore, we also wanted to test whether ASO–CCG affects R-loop formation and influences *FMR1* transcription. We used four fibroblast cell lines derived from two men with a normal range of CGG repeats (one with 20 and the other with 31 CGGs) and from two women carrying alleles with $CGG^{exp}$ (one with one normal and one $CGG^{exp}$ allele and the other with two alleles containing $CGG^{exp}$). ASO–CCG significantly increased the total level of not only *FMR1* mRNA but also *FMR1* pre-mRNA in three cell lines (Fig. 3a and Supplementary Fig. S3a). Only in cells carrying short, 20 CGG repeat tract ($CGG^{norm}$/- (1)) treatment with ASO–CCG did not affect the level of mRNA and pre-mRNA. The observed effect could be partially caused by ASO–CCG-based prevention of R-loop formation within the promoter region of *FMR1* and the subsequent increase in transcription efficiency. R-loops are structures sensitive to RNase H1, which cleaves RNA molecules within RNA/DNA duplexes and destroys these structures. Therefore, we measured the level of *FMR1* mRNA and pre-mRNA in the same fibroblasts upon the reduced level of RNase H1. In three tested cell lines treated with siRNA against RNase H1, the increase in the *FMR1* mRNA level induced by ASO–CCG was greatly diminished compared to control siRNA-treated cells, while the increase in the pre-mRNA level was no longer statistically significant (Fig. 3a and Supplementary Fig. S3a). These data suggest that ASO–CCG may enhance the transcription efficiency of the *FMR1* locus with longer CGGs by targeting R-loops and changing their thermodynamic stability.

The formation of RNase H-, RNase A-, and DNase-sensitive R-loops in the *FMR1* 5′UTR with ~100 CGGs ($rCGG_{100}$) was further confirmed in in vitro transcription system using both linear and circular DNA templates (Supplementary Fig. S3b). Our in vitro studies showed that the synthesis of $rCGG_{100}$ significantly increased in the presence of RNase H and in the presence of ASO–CCG compared to ASO-ctrl-treated samples (Fig. 3b), as was observed in cellula (Fig. 3a), which suggests that ASO–CCG influences on the transcription efficiency via modulation the stability of R-loop structure. The direct interaction between R-loops and ASO–CCG was further confirmed using fluorescently labeled ASO–CCG-Cy3 in in vitro transcription in the absence of RNase H (Fig. 3c). On the other hand, the presence of RNase H significantly reduced the amount of R-loop structures. As we expected, there was also a fraction of ASO–CCG-Cy3 which bound to a sense strand of DNA template containing $CGG^{exp}$ (Fig. 3c and Supplementary Fig. S3c). This binding may abolish the stabilizing effect of interaction between RNA and DNA structures formed by CGG repeats[8] (see Fig. 1a). Therefore, these experiments showed the ability of ASO–CCG to invade the R-loop structure, which is sensitive to RNase H, and confirmed

the positive effect of the oligonucleotide on the transcription efficiency of $rCGG_{100}$.

To further assess the degree to which the observed increase in total *FMR1* mRNA in ASO–CCG-treated FXTAS cells is related to nuclear processes (e.g., transcription or nuclear retention of mRNA) or cytoplasmic processes (e.g., mRNA stability) we performed nucleocytoplasmic fractionation. Upon ASO–CCG treatment, the nuclear level of *FMR1* mRNA was elevated only in one cell line with a normal CGG repeat number and in all $CGG^{exp}$-carrying fibroblasts, whereas its cytoplasmic level was considerably increased in all tested cell lines (Fig. 3d and Supplementary Fig. S3e). Our findings suggest that ASO–CCG affects transcription of *FMR1* alleles, especially the one carrying $CGG^{exp}$ (Fig. 3a for pre-mRNA and d for nuclear fraction). It may also impact the nuclear retention, nucleocytoplasmic transport, subcellular localization, and stability of *FMR1* mRNA. These processes jointly may lead to a pronounced increase in *FMR1* mRNA level in the cytoplasm regardless of the CGG repeat length (20–90 repeats). Since an increase in the level of *FMR1* pre-mRNA and nuclear mRNA is relatively low in ASO–CCG-treated cells we believe that the significant increase of *FMR1* mRNA in the cytoplasm is caused mainly by the elevated stability of mRNA, perhaps due to reduced efficiency of translation of both FMRpolyG and FMRP. A similar effect was previously described for many other genes[30,31].

The above-mentioned results from the artificial FXTAS cell model showed that despite the increase in $rCGG^{exp}$ level, translation of the mRNA into FMRP equivalent was reduced in the presence of ASO–CCG (Supplementary Fig. S2c). Since *FMR1* mRNA level was also increased in human fibroblasts, we next investigated whether the biosynthesis of FMRP was affected in these cells. The western blot results showed unchanged steady-state level of FMRP upon ASO–CCG treatment for five out of six cell lines (four with a normal number of CGG repeats and one with ~100 CGGs), despite of 1.5–2 times increase of its mRNA level (Fig. 3a, d, e and Supplementary Fig. S3d). Only one cell line carrying $CGG^{exp}$ showed a significantly lower level of FMRP (Fig. 3e). Our results suggest that ASO–CCG indeed reduces the efficiency of FMRP biosynthesis; however, perhaps due to the increased level of its mRNA, the steady-state level of protein is not affected.

**ASO–CCG reduces the formation of FMRpolyG inclusions and improves motor performance in an FXTAS mouse model.** The promising results obtained in cellular models of FXTAS encouraged us to evaluate the therapeutic potential of ASO–CCG in an animal model and to test whether a reduction of $CGG^{exp}$-induced toxicity would be associated with an improvement of motor functions in vivo. We selected a bitransgenic, inducible P90CGG mouse model in which the expression of a transgene containing $CGG^{exp}$ is under the control of a prion gene promoter with an expression pattern predominantly in the central nervous system and is activated by doxycycline (DOX) induction[32,33]. These mutants show a robust and controllable expression of $CGG^{exp}$ in neural cells and faithfully model FXTAS phenotypes including intranuclear inclusions and motor disturbance without signs of overt transgene toxicity or increased animal mortality, thus providing an excellent system to test the effectiveness of ASO–CCG in vivo[32,33].

The main problem with the potency of ASOs in vivo is their bioavailability. Therefore, we first investigated whether ASO–CCG could be efficiently delivered into cells in the absence of a carrier. Fluorescently labeled oligomers added directly to the cell medium were detected within the majority of cells, and an increase in the concentration of approximately twofold was needed to obtain

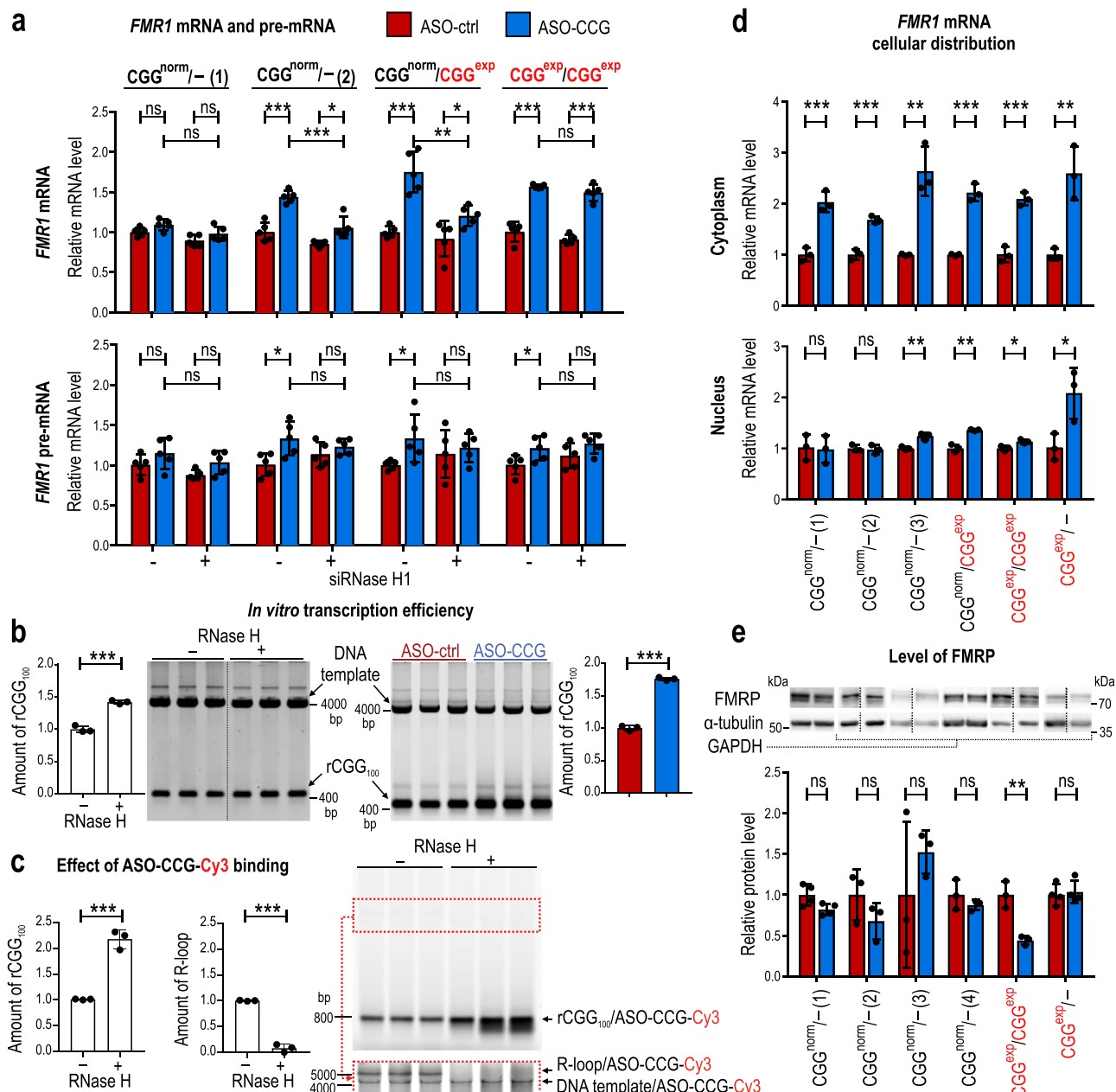

**Fig. 3 The effect of ASO–CCG on the *FMR1* pre-mRNA, mRNA, and FMRP levels in FXTAS patient-derived fibroblasts. a** Analysis of the total *FMR1* mRNA and pre-mRNA levels in fibroblasts treated with ASOs. Fibroblasts were transfected with siRNA against RNase H1 or control siRNA and with ASOs (9 or 11 nt, 200 nM). Graphs present RT-qPCR results for $N = 5$ biologically independent samples. **b** Increase in the efficiency of in vitro transcription in the presence of ASO–CCG or RNase H. $rCGG_{100}$ signal was measured for $N = 3$ independent samples. **c** Correlation between R-loop formation and efficiency of transcription of RNA containing $CGG_{100}$. Graphs present quantification of the fluorescent signal coming from fluorescently labeled ASO–CCG-Cy3 bound to either $rCGG_{100}$ or R-loop structure for $N = 3$ independent samples. The area of the gel marked with a red box was exposed to higher laser power strength and the result is presented below the main gel. DNA template was the same as in **b**, but the observed signal came only from partially single-stranded DNA in the region of CGG/CCG repeats or R-loop bound with ASO–CCG-Cy3. **d** Cellular distribution of *FMR1* mRNA in fibroblasts treated with ASOs (9 nt, 200 nM). Graphs present RT-qPCR results for the cytoplasmic and nuclear fractions for $N = 3$ biologically independent samples. Results were normalized to *GAPDH*. **e** Changes in the steady-state level of FMRP after treatment with ASO–CCG. Fibroblasts were transfected with 200 nM ASOs (9 or 11 nt). The graph presents western blot results for $N = 4$ biologically independent samples (CGG^norm/- (1) and CGG^exp/-) and $N = 3$ (other cell lines). **a–e** Graphs present means of indicated $N$ with the SDs. **a**, **b**, **d**, **e** Red bars, ASO-ctrl; blue bars, ASO–CCG. **a**, **d**, **e** The used cell lines contained one allele with a normal CGG repeat length (CGG^norm/-; lines (1), 20 CGG repeats; (2), 31 CGG repeats; (3) and (4)); one allele with a normal CGG repeat length and one allele with CGG^exp (CGG^norm/CGG^exp); two alleles with CGG^exp (CGG^exp/CGG^exp) and one allele with CGG^exp (CGG^exp/-). **b**, **c**, **d** Gels were cropped. **a–e** Two-sided, unpaired Student's *t* test; *$P < 0.05$; **$P < 0.01$; ***$P < 0.001$; ns, non-significant. Source data are provided as a Source Data file.

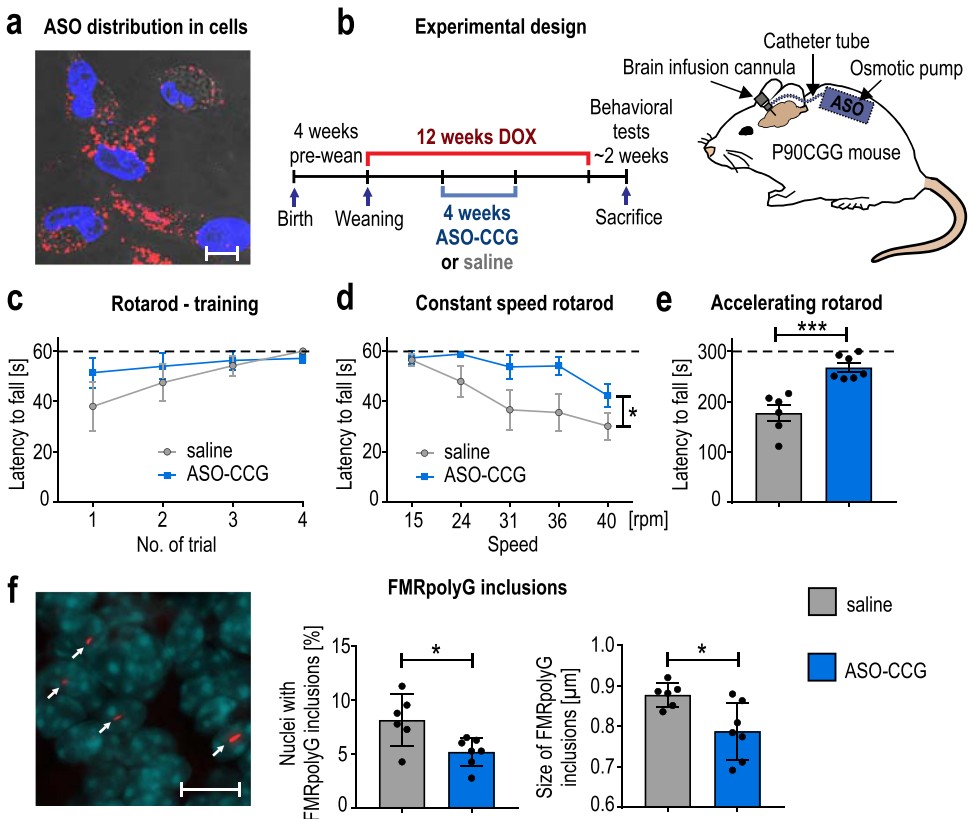

**Fig. 4 The effect of ASO–CCG on behavioral and molecular phenotypes in the FXTAS mouse model. a** Representative image showing the distribution of ASO–CCG-Cy3 (red; 200 nM) in COS7 cells. ASO entered cells in the absence of a carrier. Blue, stained nuclei; scale bar, 20 μm. The experiment was repeated two times with similar results. **b** Experimental design for DOX induction (red) and ASO–CCG delivery (blue) into P90CGG mice. The contents of the osmotic pump (ASO–CCG (11 nt) solution or saline) were delivered into the right lateral ventricle via intracerebroventricular infusion (~550 μg of ASO–CCG per mouse). **c** Training session for the rotarod test. Mice treated with saline and ASO–CCG were subjected to a 3-day rotarod test starting with a training session performed at 15 rpm constant speed (day 1). No differences were observed between groups. **d** Constant speed rotarod test. On day 2, the above-mentioned mice were subjected to a constant speed rotarod test at different speeds (x axis; rpm, revolution per minute). The ASO–CCG-treated group stayed longer on the rod, $P = 0.0323$. **e** Accelerating rotarod test. On day 3, mice were subjected to an accelerating rotarod test (4–40 rpm ramp). The ASO-CCG-treated group showed better performance on this test (longer latency to fall), $P = 0.0003$. Dashed line, maximum cutoff **c**, **d** 60 s, **e** 300 s. **f** Quantification of FMRpolyG foci in P90CGG mice. A representative image of FMRpolyG staining, sections containing cerebellar lobule X of P90CGG mice was stained using the 8FM antibody specific for FMRpolyG (scale bar, 10 μm). For the ASO–CCG-treated group, the percentage of the nuclei with positive staining for FMRpolyG (red) was lower ($P = 0.0161$) and the size of the inclusions they contain (indicated by a white arrow) were smaller ($P = 0.0146$) than that of the saline-treated group. Graphs present means with SDs. **a**, **f** Images were pseudo-colored and merged. **c**, **d** Two-way repeated-measures ANOVA (treatment effect): *$P < 0.05$. **e**, **f** Two-sided, unpaired Student's $t$ test: *$P < 0.05$; ***$P < 0.001$. **c**–**e** Graphs present means with SEMs. **c**–**f** Gray dots/bars, saline-treated animals ($N = 6$); blue squares/bars, ASO–CCG treated animals ($N = 7$). Source data are provided as a Source Data file.

an inhibitory effect on RAN translation comparable to that of transfection with lipophilic carriers (Fig. 4a and Supplementary Fig. S4a). Therefore, we assumed that unassisted uptake of ASO–CCG into neurons in vivo was feasible when delivered via intracerebroventricular infusion.

To check the effect of ASO–CCG on FXTAS-specific symptoms in P90CGG mice, we designed a specific treatment procedure featuring an intracerebroventricular infusion of either ASO–CCG or saline as a control. Our previous study showed that 12 weeks of DOX-induced expression of $CGG^{exp}$ is sufficient to obtain a characteristic FXTAS-specific phenotype, e.g., the formation of FMRpolyG inclusions in neural cells and motor deficits, that is stable after cessation of transgene induction. Therefore, we adopted a $4 + 4 + 4$ treatment schedule (Fig. 4b), with ASO–CCG infusion starting after 4 weeks of DOX induction. In the ensuing 4 weeks, mice received both DOX and ASO–CCG followed by 4 additional weeks of the only DOX. Then, animals were subjected to motor behavior tests, and, finally, about 17 days after the last DOX exposure brain tissues were collected and molecular assays were performed.

We used rotarod tests to assess the motor performance of treated animals. In our previous study, while P90CGG DOX-control mice performed close to maximum cutoff levels on the rotating rod, significant deficits were present in the P90CGG mice upon 12 weeks of DOX induction both on the constant high speed and the accelerating rotarod. Moreover, a shorter 8-week transgene expression was not sufficient to drive these deficits[33]. We therefore now, in the ASO-treated FXTAS model under 12 weeks of DOX induction and comparable test conditions performed a series of rotarod tests over three days comprising of a training session, a constant speed test session, and an acceleration ramp test session. In the training session, at low speed, ASO-treated mice showed no differences in the latency to fall off the rod compared to saline-treated mice (Fig. 4c), suggesting a comparable acclimation level in both groups at the start of the testing session on day 2. However, in the constant speed rotarod test, the performance of ASO-treated mice was significantly better than that of their saline-treated counterparts especially at higher speeds (Fig. 4d). The performance of ASO-treated mice was also better on day 3, as they remained on the accelerating rod

significantly longer than saline-treated mice (Fig. 4e). The rotarod test results were not affected by differences in the body weights of the tested mice (Supplementary Fig. S4b).

Then, we investigated the molecular mechanism underlying the behavioral improvements following ASO–CCG treatment. Intranuclear inclusions, the hallmark of FXTAS pathology, are considered to reflect a neurotoxic action, and we previously reported that large numbers of these inclusions were present in cerebellar lobule X[32,33]. We quantified the number of intranuclear inclusions in the granule cell layer of this brain structure using an antibody specific for the FMRpolyG peptide (Fig. 4f, left and Supplementary Fig. S4c). Not only the percentage of nuclei with positive staining for FMRpolyG but also the average size of the inclusions was significantly decreased in ASO-treated brains (Fig. 4f, right).

**ASO–CCG ameliorates the transcriptome changes of FXTAS mouse brain.** To gain deeper insight into the molecular changes induced by ASO–CCG treatment, we conducted RNA-seq analyses of the striatum, one of the brain structures with the highest level of transgene expression[32]. Approximately 50–70 million reads were obtained for each sample, and differential analysis of ~17,000 genes was performed for three groups of animals: bitransgenic P90CGG mice expressing CGG$^{exp}$ and treated with either (1) ASO–CCG or (2) saline and (3) monotransgenic mice harboring the transgene containing CGG$^{exp}$ but not expressing it (the control group; all mice were treated with DOX). The expression of 417 genes was significantly changed (adjusted $P$ value <0.05) in P90CGG mice treated with saline compared to control mice (Supplementary Data S1). These CGG$^{exp}$-dependent changes can be considered FXTAS-related. Therefore, we expected that treatment with ASO–CCG could restore the expression of at least some of these genes toward the level observed in the control group. Indeed, we found a significant global correction of the expression of FXTAS-related genes in ASO–CCG-treated mice, as evidenced by the negative Pearson correlation ($r = -0.75$; $P < 0.001$) (Fig. 5a, upper panel and Supplementary Fig. S5). The expression level of 57 of the 417 genes was completely restored, and the correction of most of the others was substantial. Then, we performed gene ontology analysis to obtain better insight into the functions of abnormally expressed FXTAS-related genes (Supplementary Data S2). Among the downregulated genes, genes involved in translation (mostly structural constituents of the ribosome), amide biosynthesis, and oxidation-reduction processes were significantly enriched (adjusted $P$ value <0.001). The enriched pathways were nonsense-mediated RNA decay, respiratory electron transport, ribosomal scanning, and start codon recognition (Fig. 5a, gray bars). On the other hand, among the upregulated genes, genes linked with nervous system development and playing a role in transcription coregulator activity were significantly enriched. Importantly, all of these gene ontology categories were mainly unchanged in ASO–CCG-treated mice compared to control mice (Fig. 5a, blue bars). These results again suggest significant rescue of the molecular phenotype observed at the transcriptome level after ASO–CCG treatment.

Next, to examine the off-target effects of ASO–CCG in vivo, we performed RNA-seq analysis of both the striatum as well as the cortex, in which the expression of CGG$^{exp}$ is very low but the concentration of ASO–CCG is expected to be high due to the direct contact of this structure with the cerebrospinal fluid where the ASO was infused into. Among the ~16,000 analyzed genes, ~150 contain at least six uninterrupted CGG repeats in exonic sequences, mainly in the 5′UTRs. On a global scale, the expression of this group of genes was significantly elevated in

ASO-treated mice compared to saline-treated mice (Fig. 5b and Supplementary Fig. S6a, b; Supplementary Data S3 and S4), confirming our previous observation in cell-based models of increased levels of mRNAs containing CGG repeats. Gene ontology analysis showed slight but statistically significant enrichment of immune system-related genes, suggesting that ASO–CCG induces an immune response in the brain cortex (Supplementary Table S1).

Therefore, observations coming from RNA-seq analysis pushed us to study both the off-target effect and immune system-related toxicity of ASO–CCG on a protein level. As potential off-targets, we chose genes containing CGG repeats in their 5′UTRs and exhibiting an elevated mRNA level in the ASO–CCG-treated brain tissue, according to the RNA-seq data. Protein products of these genes were analyzed using western blot. The level of none of them was changed in a result of ASO treatment (Fig. 5c). Moreover, we checked the level of these proteins in two fibroblast cell lines with normal and expanded CGG repeats in *FMR1* gene. We found that the level of only two out of four tested proteins, namely QKI (Protein quaking) and VKORC1L1 (Vitamin K epoxide reductase complex subunit 1-like protein 1), was significantly reduced, but in solely one ASO–CCG-treated cell line (Supplementary Fig. S6c, d). Treatment with ASO–CCG in either type of cell did not exert any effect on the level of DAZAP1 (DAZ-associated protein 1), whose mRNAs contain 14 CCG repeats within the 5′UTR.

Immune system-related toxicity markers were analyzed on the hippocampus and cortex derived from saline or ASO–CCG-treated mice. Western blot revealed that a marker of gliosis, glial fibrillary acidic protein (GFAP) was present at similar levels in both groups of mice, in spite of an increase at mRNA levels detected from ASO–CCG-treated brains (Fig. 5d, e). Since the transformation of full-length GFAP into shorter form is considered as a sign of glial injury[34] we also quantified GFAP's proteolytic fragment level and observed an increase, however, not statistically significant. Similarly, the increase at the mRNA levels of CD68 (macrosialin), a marker of activated microglia, was not observed at the protein level. The protein levels of allograft inflammatory factor 1 (AIF1), another marker of microglia, tumor necrosis factor α (TNF-α), a marker of inflammation, as well as protein phosphatase 1 regulatory subunit 1B (PPP1R1B), a marker of medium spiny neurons, were also unchanged in ASO–CCG-treated brains (Fig. 5d, e and Supplementary Fig. S6e, f). Overall, our investigation of potential toxicity markers shows no evidence of prominent adverse effects related to ASO–CCG treatment.

## Discussion

Antisense strategies using steric blockers are being widely tested as therapeutics for a range of human diseases, including neurodegenerative disorders[35]. This class of ASO binds to its targeted RNA sequence and modifies the maturation or metabolism of the targeted RNA without inducing transcript degradation. Recently, two ASO steric blockers that modify the splicing of specific exons were approved for the treatment of Duchenne muscular dystrophy (DMD) and spinal muscular atrophy (SMA)[36,37]. Short ASOs composed exclusively or predominantly of LNA units were successfully used to repress miRNA activity in animal models, and their activity against the hepatitis C virus is being tested in clinical trials[38,39]. The advantages of using LNAs are the low toxicity, resistance to cellular nucleases, and extremely high thermodynamic stability of the duplex formed with complementary RNA. This allows the use of very short ASO sequences that can penetrate into cells and tissues in the absence of specific carriers via a receptor-based endocytosis mechanism known as

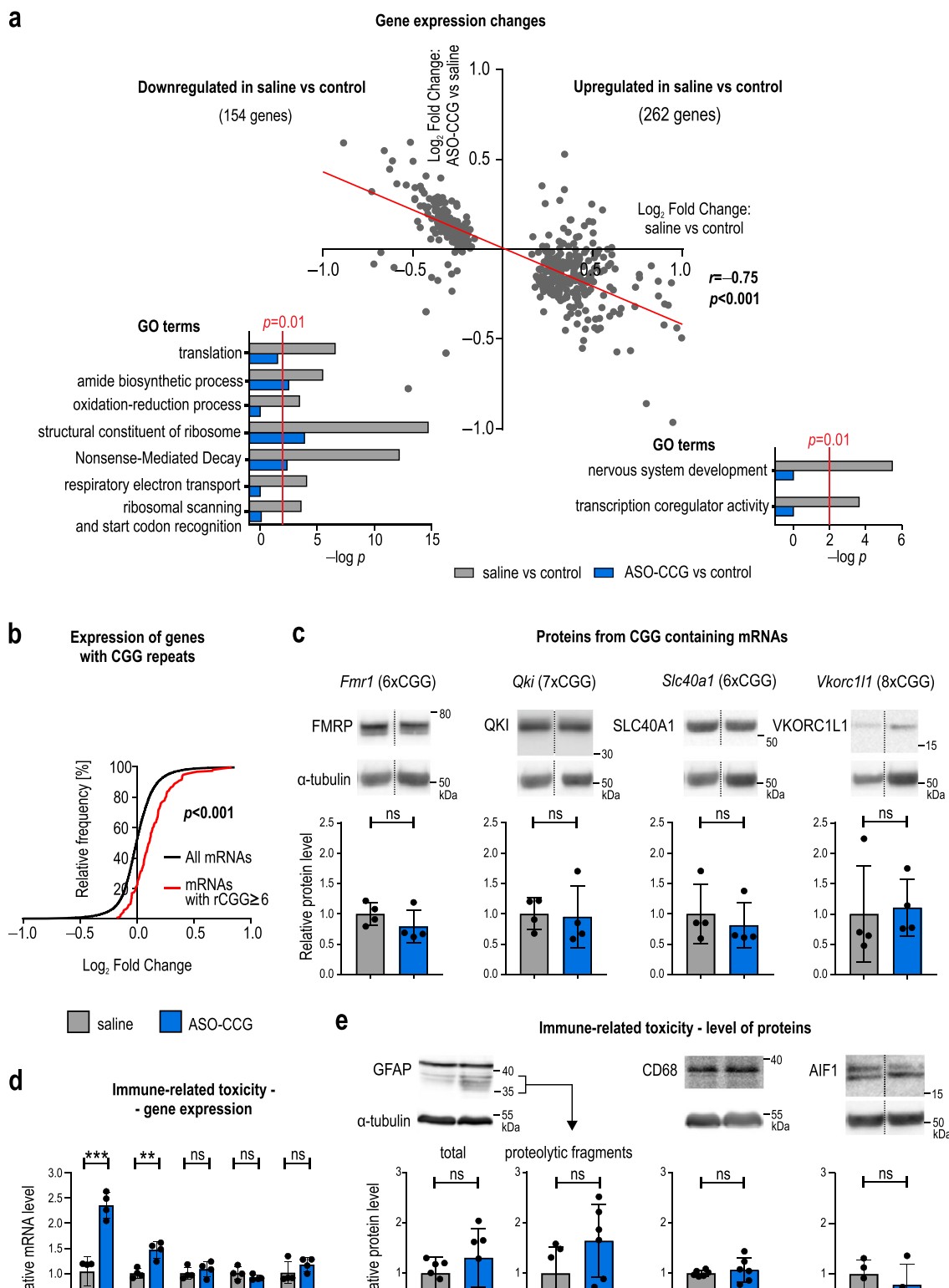

gymnosis[39]. We showed that ASO–CCG also exhibits these features (Fig. 4a). Moreover, our results suggest that ASOs in which the number of LNA units is a multiple of 3 (6, 9, 12, etc.) bind even more tightly to targeted triplet repeat RNAs because of the stabilizing effect of stacking interactions between the bases of consecutive ASOs (Fig. 1b and Supplementary Fig. S2b). We believe that this observation may also help to design antisense therapeutic strategies for other 3–6 nucleotide repeat expansion diseases.

Here, we showed that ASO–CCG affects multiple stages of mutant *FMR1* gene expression, e.g., transcription, nuclear metabolism, and translation. The increased levels of target mRNAs containing both short and expanded CGG repeats upon ASO–CCG treatment suggest the involvement of this compound

**Fig. 5 Beneficial and adverse effects of ASO–CCG (11 nt) at a molecular level in an FXTAS mouse model. a, b** DE, results based on differential gene expression analysis of the RNA-seq data. adj.P.Val, $P$ value generated using moderated $t$-statistic and adjusted for multiple testing using Benjamini–Hochberg's method. **a** Plot, DE for striatum of control, saline- or ASO–CCG-treated P90CGG mice ($N = 4$ per group). On the $x$ axis, the $\log_2$ fold change in the expression levels of the 416 genes (one outlier gene, *Gm20425*, was excluded from analysis), that were significantly changed (adj.P.Val < 0.05) in saline-treated P90CGG mice compared to control mice is plotted. On the $y$ axis, the $\log_2$ fold change in the gene expression levels of the same genes compared between ASO–CCG- and the saline-treated group is plotted. A general restoration of the expression levels towards the levels in control mice was observed in the ASO–CCG-treated group (red line, negative Pearson correlation with $r = -0.75$ and $P < 0.001$). Bar graphs, gene ontology analysis of genes extracted from DE for striatal tissue according to the following parameters: average expression value >1 and adj.P.Val ≤ 0.05. Significantly upregulated and downregulated genes were analyzed for the saline vs control (gray) and ASO–CCG vs control groups (blue). Red line, $P = 0.01$ in Fisher's exact test with Bonferroni correction for multiple testing. **b** Analysis of expression changes in genes containing CGG repeats in ASO–CCG-treated P90CGG mice. DE for the cortex of saline- or ASO–CCG-treated P90CGG mice ($N = 4$ for each group). A global comparison of gene expression between these two groups of mice showed a significant increase in the expression of 22 genes containing at least 6 CGG repeats (red line) compared to all analyzed genes (black line; $P < 0.001$, two-tailed Mann–Whitney test). Moreover, in the list of genes showing significant expression changes between these two groups (adj.P.Val < 0.05), genes containing ≥ 6 CGG repeats were enriched 13-fold relative to the expected level. **c** Steady-state level of proteins encoded by genes carrying short CGG repeats in 5′UTRs in the cortex of saline- and ASO–CCG-treated P90CGG mice. $N = 4$ animals. The CGG repeat length is specified for each gene. **d** RT-qPCR of immune system-related genes in the cortex of P90CGG mice. $N = 4$ animals. **e** Steady-state level of immune system-related toxicity markers in P90CGG mice. Hippocampus: GFAP, CD68, $N = 6$ animals. Samples were derived from the same experiment and processed in parallel on different gels. Cortex: AIF1, $N = 4$ animals. **c, e** Western blot. All blots were cropped. **c–e** Gray bars, saline-; blue bars, ASO–CCG-treated animals. Graphs present means of indicated $N$, with the SDs. Two-tailed unpaired Student's $t$ test, **$P < 0.01$; ***$P < 0.001$; $P > 0.05$, ns, non-significant. **a–d** Source data are provided as a Source Data file.

in cotranscriptional processes, such as R-loop formation (Fig. 3a–c and Supplementary Fig. S3a, c). A previous study suggested that CGG[exp] within the *FMR1* locus either increase the formation efficiency of R-loops or positively affect their thermodynamic stability[6,8]. Here, we showed that the presence of RNase H-sensitive R-loops within the *FMR1* locus decreases *FMR1* transcription both in vitro and in vivo (Fig. 3a–c), perhaps due to the formation of a steric barrier for RNA polymerase. Furthermore, we found that ASO–CCG reduces R-loop formation and/or their stability and significantly increases *FMR1* transcription. The ASO–CCG-induced increase in *FMR1* pre-mRNA and mRNA is significantly milder in FXTAS-derived cells with RNase H1 deficiency, which generally reduces the negative effect of R-loops on transcription (Fig. 3a, b and Supplementary Fig. S3a). R-loops within CGG[exp] loci may cause DNA breakage and induce the DNA damage response in FXTAS cells[6], possibly leading to neuronal death. Therefore, ASO–CCG may also exert a protective effect against double-stranded DNA breaks by inhibiting R-loop formation and/or reducing their durability. Beside cotranscriptional events, several post-transcriptional processes may also induce an increase in the *FMR1* mRNA level due to ASO–CCG treatment. One potential mechanism could be increased efficiency of nucleocytoplasmic transport, what we previously showed for mRNA containing CUG repeat expansion (rCUG[exp]) after treating muscles of myotonic dystrophy mice with short LNA-composed ASOs targeting CUG repeats[40,41]. Other mechanisms could be linked to the increased stability of mRNA. According to recent findings, slowing down the ribosome movements or lowering the density of ribosomes on *FMR1* mRNA caused by ASO–CCG binding (Fig. 2e) may increase its half-life, an observation that was previously reported for hundreds of mRNAs[30,31].

*FMR1* transcripts containing CGG[exp] have been found in nuclear inclusions both in the brains of patients with FXTAS and in cell models[9,13]. Many proteins required for RNA maturation, transport, stability, and translation are efficiently sequestered by binding directly or indirectly to rCGG[exp][10–16]. We showed that ASO–CCG led to a general increase in miRNA levels and corrected aberrant splicing events in cells overexpressing CGG[exp], which suggests that ASO–CCG released DGCR8 and SAM68 from sequestration (Fig. 1e, f and Supplementary Fig. S1f). The activity of other proteins affected by binding to rCGG[exp] may also be restored by ASO–CCG. Previously, we applied a similar

strategy in a mouse model of myotonic dystrophy, in which ASO blockers targeting CUG[exp] corrected the alternative splicing of hundreds of genes due to the release of MBNLs (muscleblind-like splicing regulators) from pathogenic sequestration on rCUG[exp][40,41]. RNAs with the expansion of other GC-rich tandem repeats, such as (rGGGGCC)n or (rCUG)n, were reported to form gel-like nuclear inclusions in a repeat length-dependent manner[42,43]. These processes can enhance the sequestration efficiency of RBPs. Therefore, abolishing the interaction of such proteins with rCGG[exp] and also the interactions between different rCGG[exp] molecules should globally reduce RNA-mediated toxicity.

The toxicity of FMRpolyG was proven in *Drosophila* and mouse models[18,19,44]. Mice expressing the full-length 5′UTR of *FMR1* containing CGG[exp] show Purkinje cell loss, locomotor impairment and decreased viability, while mice in which the sequence controlling RAN translation initiation located upstream of CGG[exp] was deleted do not exhibit such symptoms[19]. Recently, the expression of rCGG[exp] was shown to inhibit global translation and to activate the cellular stress response[45]. In turn, stress-induced phosphorylation of eIF2α enhances RAN translation[45]. These data suggest that RAN translation is part of a feed-forward loop that may lead to neurodegeneration[45]. As ASO–CCG decreases the level of toxic FMRpolyG production (Figs. 2b, c, 4f and Supplementary Fig. S2a–c), it may significantly slow disease progression. Moreover, the results of our transcriptome-based studies showed that the levels of genes responsible for translation were rescued in ASO–CCG-treated FXTAS mice (Fig. 5a). The decrease in FMRpolyG biosynthesis from mutant *FMR1* mRNA may occur at different stages of translation, e.g., initiation or elongation. By binding individually to rCGG[exp], ASO–CCGs can prevent either the formation of the initiation complex during the scanning phase of the 43S ribosome subunit or the stalling and dissociation of the 80S ribosome. Our sucrose-gradient fractionation experiments, which showed an increase in the level of CGG[exp]-carrying mRNAs in the free mRNA and monoribosome fractions after ASO–CCG treatment suggest that ASO–CCG can block the translation of both FMRP and FMRpolyG (Fig. 2e). A similar strategy for suppressing the translation of toxic proteins was applied in cell models of Huntington disease and spinocerebellar ataxia type 3 (SCA3), neurodegenerative disorders caused by the dominant effect of trinucleotide repeat expansions[46]. Targeting the CAG repeats with ASOs in a cellular model

selectively decreased the biosynthesis of proteins from mutant mRNA carrying expanded CAG repeats in an open reading frame but not of normal huntingtin and ataxin-3[46]. Moreover, ASO targeting the initiation site of rCGG[exp] RAN translation in *FMR1* mRNA was able to reduce the level of toxic FMRpolyG and at the same time increase FMRP in cell models[24]. Our data show that ASO–CCG reduces the level of FMRpolyG and the number of FMRpolyG foci in cell models and in vivo (Figs. 2b, c, 4f and Supplementary Fig. S2a–c), but it may also reduce the steady-state level of FMRP (Fig. 3e and Supplementary Fig. 2c). This observation should be carefully considered when planning potential FXTAS therapies as induction of FMRP deficiency in adults may lead to impairment of neuronal functions[47]. Thus, determining the appropriate dose of ASO–CCG to ensure both the preservation of sufficient FMRP levels and a beneficial reduction in the pathogenic processes causing FXTAS should be the goal of future studies.

In the P90CGG DOX-induced mouse model, the transgene containing human 5′UTR of *FMR1* with CGG[exp] outside of the context of murine *Fmr1* is controlled by a prion gene promoter, with the highest expression in the hippocampus, striatum, and cerebellum, particularly in lobule X. This phenotype is accompanied by the greatest number of intranuclear FMRpolyG inclusions in the above-mentioned brain regions and by behavioral impairments connected with their functions[32,33]. The time course of phenotype induction in this model has been carefully established. After 12 weeks of DOX induction, mice exhibited a phenotype of decreased motor performance, while after 8 weeks of exposure, most mice remained asymptomatic[33]. No spontaneous recovery of motor functions and no decrease in the number of inclusions in cerebellar lobule X could be observed once these phenotypes have developed[33], demonstrating the need for early intervention. On the other hand, the stability of the phenotype even after cessation of the 12-week transgene induction, as well as the absence of an overt toxicity phenotype or increased mortality in these mice, provide an ideal framework for testing the in vivo effectiveness of the ASO treatment. Therefore, we adopted a 4 + 4 + 4 schedule (Fig. 4b) with constant ASO–CCG infusion over 4 weeks starting at week 4 of DOX administration (for a total of 12 weeks of DOX induction). Thereby, it was possible to reduce the formation of FMRpolyG inclusions in cerebellar lobule X along with significant improvement of motor performance compared to saline-treated controls (Fig. 4d–f). However, it is not clear at this point whether the improvement in motor performance is the direct result of the reduction in the number or size of inclusions in lobule X or a correlate. On the other hand, this reduction is a strong indication of the potency of the ASO–CCG because lobule X is a region, where the transgene expression is particularly high in our P90CGG model. An interesting question to be addressed is whether this intervention is sufficient to provide lasting protection or merely slows down the pathogenic processes. Based on the encouraging findings of our current study we believe that further preclinical assessment of ASO–CCG that takes into consideration potential long-lasting effects of different treatment protocols, as well as the influence of age and treatment onset, is warranted in P90CGG mutants and other FXTAS models.

Cerebellar lobule X has been widely used to monitor inclusion formation in the P90CGG model. However, it should be considered that other cerebellar regions as well as basal ganglia also show high transgene expression and are involved in motor performance and disease pathology. Therefore, to address further functional changes, we have investigated the therapeutic potential of ASO–CCG at the transcriptome level from striatum tissue. In ASO–CCG-treated mice, the expression levels of FXTAS-related genes were similar to the expression levels observed in control

animals (Fig. 5a). One example is the group of genes responsible for the regulation of translation and mitochondrial function. In FXTAS cell models these processes are impaired[45,48]. Treatment with ASO–CCG also increased the level of mRNAs carrying short CGG repeats located mostly in the 5′UTR, but this effect was statistically significant for only ~14% of such transcripts (Fig. 5b). However, upon ASO–CCG treatment, the level of protein products of selected genes was not affected in mouse brains and slightly reduced in human cells for only two such genes (Fig. 5c and Supplementary Fig. S6c, d). Another adverse effect of ASO–CCG observed at the mRNA level was the induction of a few genes controlling the immune response (Fig. 5d and Supplementary Table S1). Nevertheless, the induction of these genes did not result in detectable changes at the steady-state protein levels in the ASO–CCG treated brains (Fig. 5e and Supplementary Fig. S6e).

In summary, we showed that short oligonucleotide steric blockers composed exclusively of LNA units targeting rCGG[exp] can affect multiple stages of mutant *FMR1* gene expression. ASO–CCG can interfere with cotranscriptional R-loop formation and DNA damage, RNA-mediated toxicity, and toxic protein biosynthesis, indicating its compelling therapeutic potential for the treatment of FXTAS, FXPOI, and other diseases caused by CGG[exp] [49]. The potency of ASO–CCG was clearly manifested in the amelioration of behavioral and molecular phenotypes in the FXTAS mouse model. Importantly, the therapeutic strategy presented in this work, which relies on targeting RNA containing expanded microsatellite repeats with ASO steric blockers, could also be used to treat other diseases with similar etiology via cerebrospinal fluid delivery.

## Methods

**Antisense oligonucleotides**. The 9 or 11-nucleotide-long ASOs were composed of 8 or 10 LNA units, respectively, and a 2′-O-Me unit at 3′ end. All LNA positions were phosphorothioated. The sequence of ASO–CCGs is 5′-CCGCCGCCGCC-3′ (11 nucleotides long) or 5′-CCGCCGCCG-3′ (9 nucleotides long), the sequence of ASO-ctrl is 5′-TTGAACATAAG-3′ (11 nucleotides long) or 5′-TGAACATAA-3′ (9 nucleotides long) and ASO–CCG-Cy3 is 9-nucleotide-long oligonucleotide 5′-CCGCCGCCG-3′ with Cy3 modification at a 5′ end. Toxic ASO used as a positive control in viability and apoptosis assays is 15-nucleotide-long ASO composed of LNA units (bolded) located at 5′ and 3′ ends and DNA nucleotides in the central core. All positions were phosphorothioated. The sequence of toxic ASO is 5′-**CCG**CCGCCGCCG**CCG**-3′. ASOs were synthesized and HPLC purified by Kaneka Eurogentec.

**Genetic constructs**. Genetic constructs containing CGG repeats used in this work were previously described: (20×CGG, 60×CGG)[13], (100×CGG)[19] and (100×CGG-mCherry)[23]. The 20×CGG and 60×CGG contain bare CGG repeats without flanking sequences of *FMR1* gene. The 100×CGG contains 5′UTR sequence of *FMR1* gene with ~100 CGG repeats and located downstream GFP sequence and the product of its overexpression is fusion FMRpolyG-GPF protein. The 100×CGG-mCherry contains 5′UTR sequence of *FMR1* gene with ~100 CGG repeats and located downstream mCherry sequence and products of its expression are FMRpolyG and mCherry which translation starts from FMRP-specific start codon (FMRP-N-mCherry). MB1-40-mCherry construct[50] was used in experiments based on the detection of fluorescent proteins after SDS-PAGE separation (FL-SDS-PAGE method, referred to as reference mCherry construct). MBNL3-39-GFP construct[50] was used in flow cytometry (referred as control GFP construct). *SMN2* minigene (pCI SMN2) was a gift from Elliot Androphy (Addgene plasmid # 72287; http://n2t.net/addgene:72287; RRID:Addgene_72287)[51]. The SAM68-independent control, *Atp2a1* minigene, was described in ref. [52].

**Electrophoretic mobility shift assay (EMSA)**. The rCGG$_{20}$ was a gift from W. Krzyżosiak (The Institute of Bioorganic Chemistry, Polish Academy of Sciences). 5′ radiolabeling of rCGG$_{20}$ was performed as previously described[52] followed by 1-min denaturation step. EMSA was carried out by incubating 5′ radiolabeled rCGG$_{20}$ (1 nM) with 9 or 11-nucleotide-long ASO–CCG of indicated concentrations (ranging from 0 to 500 nM). Reactions were performed in a volume of 10 μl, in 1× buffer E (50 mM NaCl, 50 mM KCl, 50 mM TRIS-HCl, 1 mM MgCl$_2$, pH 8.0) and incubated at 37 °C for 15 min. The samples were run on a native 8% PA gel in 0.5× TBE at 100 V for 1 h. The gel was subsequently dried, and the signal was detected overnight, visualized on IP through FLA-1500 (FujiFilm) and quantified

using Multi Gauge 3.0 software (FujiFilm). $K_d$ measurement of the rCGG$_{20}$/ASO–CCG interaction was based on the percentage of free rCGG$_{20}$ using one phase decay curve in GraphPad program, represented by an equation $Y = (Y0 - Plateau)*exp(-K*X) + Plateau$.

**Cell culture and transfection**. COS7 cells were grown in a high glucose DMEM medium with L-glutamine (Lonza), 10% fetal bovine serum (Merck), 1% antibiotic/antimycotic (Merck) at 37 °C in 5% $CO_2$. Fibroblasts (CGG$^{norm}$/- (2), C0603; CGG$^{norm}$/CGG$^{exp}$, FX11-02; CGG$^{exp}$/CGG$^{exp}$, WC26; CGG$^{norm}$/- (1), C6; CGG$^{exp}$/-, F3; CGG$^{norm}$/- (3), GM23963; CGG$^{norm}$/- (4), GM04033) were previously described[53,54] and grown in EMEM medium (Lonza), 15% fetal bovine serum (Merck), 1% MEM nonessential amino acids (Thermo Fisher Scientific) and 1% antibiotic/antimycotic (Merck) at 37 °C in 5% $CO_2$. For transfection with genetic constructs, COS7 cells were plated on the appropriate cell culture vessels. Cells were transfected 2 h from plating with genetic constructs at ~80% confluency. After 3–4 h, cells were transfected with ASOs. Fibroblasts were plated on appropriate cell culture vessels and transfected at ~80% confluency. In all experiments, ASOs were denatured before transfection for 30 s at 95 °C and chilled on ice. All transfections were made with the use of Lipofectamine 3000 (Thermo Fisher Scientific) according to the manufacturer's instructions. Specific conditions for all experiments are stated below.

**Monitoring of cell viability in real time**. For cell viability assay, COS7 cells were plated on a 96-well plate in 150 μl of media. The next day, cells were transfected with 9-nucleotide-long ASOs and toxic ASO at 200 nM concentration. After 6 h transfection mixture was removed and replaced with 150 μl of media containing real-time cell viability reagents (Real-Time-Glo MT Cell Viability Assay, Promega). The luminescence signal was read on a Tecan Spark reader set after 7, 14, 23, 32, 46, and 64 h. At the time of transfection cell, confluency was about 40% and reached 90% in mock control after 52 h of culture.

**Apoptosis assay**. For apoptosis assay, COS7 cells were plated in a 24-well plate and transfected 5 h from plating with 9-nucleotide-long ASOs and toxic ASO at 200 nM concentration. As additional positive control cells treated with 50 μM CCCP (carbonyl cyanide 3-chlorophenylhydrazone). After 24 h, apoptosis of the cells was tested with Guava® MitoDamage Kit (Luminex) according to the manufacturer's protocol. Samples were analyzed on a guava easyCyte HT flow cytometer and guavaSoft™ software, version 3.1.1 (Luminex) using parameters and gating strategy assigned to dedicated MitoDamage Kit software template. Cells with lower MitoSense Red fluorescence were considered as positive for apoptosis (Supplementary Fig. S7).

**miRNA-level quantification**. For mature miRNA-level quantification after treatment with ASOs, we used COS7 cells, 12-well plates, 1 μg of 100×CGG construct per well, and 9-nucleotide-long ASOs at 200 nM concentration. For mature miRNA level quantification upon expression of normal or expanded CGG repeats, we used COS7 cells, 24-well plate, and 500 ng of 20×CGG or 60×CGG or 100×CGG construct per well. After 48 h or 72 h, cells were harvested using TRI Reagent (Merck) and the total RNA was isolated according to the manufacturer's instructions or with the use of Total RNA Zol-Out™ D (A&A Biotechnology). RNA quantity and quality were checked with the use of a spectrophotometer (DeNovix). Complementary DNA (cDNA) was synthesized in a coupled poly-adenylation reverse-transcription reaction by using 2 μg of the total RNA for 1 h at 37 °C in RT buffer (10 mM Tris-HCl pH 8.0, 75 mM KCl, 10 mM DTT, 70 mM MgCl$_2$, 20 U RNasin, 2.5 mM of all four deoxynucleoside triphosphates, and of rATP, and 800 ng of anchored oligo(dT) primer with a sequence 5′-GTG CAG GGT CCG AGG TTC AAC TAT AGG TTT TTT TTT TTT TTT TTT TTT TTT VN-3′ supplemented with 200 U of Superscript III reverse transcriptase (Thermo Fisher Scientific) and 5 U of E. coli poly(A) polymerase (PAP; New England Biolabs). Reactions were heat-inactivated for 10 min at 85 °C. Then, 2 μl of the 9× diluted cDNA template was used for each qPCR with a universal reverse primer (UR) with sequence identical to the part of anchored oligo(dT) primer (Supplementary Table S1) and a probe-specific forward primer (600 nM each). qRT-PCR was performed using the Power SYBR Green PCR Master Mix (Thermo Fisher Scientific), and samples were run in technical triplicates on a 7900HT Fast Real-Time PCR instrument. Ct values were normalized against the internal controls, U6. Fold differences in expression level were calculated according to the $2^{-\Delta\Delta Ct}$ method. All primers are listed in Supplementary Table S2.

**Alternative splicing analysis**. For SAM68-dependent alternative splicing analysis upon CGG$^{exp}$ expression, we used COS7 cells, 24-well plate, 300 ng of either 20×GG or 60×CGG or 100×CGG construct, 100 ng of SMN2 minigene and 100 ng of Atp2a1 minigene per well and 11-nucleotide-long ASOs at 100 or 200 nM concentration. After 48 h, cells were harvested using TRI Reagent (Merck), and the total RNA was isolated according to the manufacturer's instructions. cDNA was synthesized with the use of GoScript™ Reverse Transcriptase System (Promega) and random primers (Promega) according to the manufacturer's instructions. PCR was performed with GoTaq DNA Polymerase and primers listed in Supplementary Table S2. Sequences of SMN2_F and SMN2_R primers were obtained from[13]. PCR

products were separated in 1–2% agarose gel with ethidium bromide. Images were captured with the use of G:Box EF2 (Syngene). Signals of PCR products were assessed with the use of GeneTools (Syngene). Percent of alternative exon inclusion (PSI, the percent of alternative exon spliced in) was calculated based on signals of two bands according to the following formula (isoform with included exon*100)/(isoforms with included exon + excluded exon).

**Analysis of ASO–CCG cellular uptake**. COS7 cells were plated on a 96-well plate for epifluorescence microscopy or on μ-Slide 8 Well (Ibidi) for confocal microscopy. ASO–CCG-Cy3 was introduced to cells via transfection or added directly to the cell at 200 nM 3–4 h after plating. For epifluorescence microscopy after 24 h cell medium was replaced with PBS. Images were taken with Axio Observer.Z1 microscope equipped with AxioCam MRm camera, filter set excitation 540–552, emission 575–640, A-Plan ×10/0.25 Ph1 objective (Zeiss), and AxioVs40 module. For confocal microscopy, Hoechst 33342 (Thermo Fisher Scientific) was added to cell medium to final concentration 5 μg/ml 24 h post ASO delivery, and cells were incubated for 15 min at 37 °C. Images were captured with the use of a Nikon A1Rsi confocal microscope with Nikon Apo ×40 WI λ S DIC N2 objective. Hoechst 33342 and Cy3 were excited with 405 nm diode laser and 561 nm pumped-diode laser, respectively. For detection dichroic mirror 405/488/561 nm with spectral filters, 450/50 nm and 595/50 nm were used.

**Flow cytometry**. For assessment of total FMRpolyG-GFP, we used COS7 cells, 48-well plate 250 ng of 100×CGG or control GFP per well, and 11-nucleotide-long ASOs at 100 or 200 nM. After 48 h, cells were analyzed with the use of flow cytometry protocol described in detail in ref.[23]. Shortly, cells were trypsinized and suspended in warm PBS. Propidium iodide (PI) at final concentration 1 μg/ml was added to stain dead cells which were next excluded from analysis. GFP and PI signal in cells was analyzed with guava easyCyte™ HT flow cytometer and guavaSoft™ software, version 3.1.1 Luminex). The gating strategy is presented in Supplementary Fig. S8.

**Microscopic analysis of FMRpolyG-GFP inclusions**. For assessment of FMRpolyG-GFP inclusions, we used COS7, 96-well plate, 125 ng of 100×CGG construct per well and 11-nucleotide-long ASOs at 100 or 200 nM concentration. After 48 h, the culture medium was removed and cells were washed in warm PBS to remove dead cells. Next, cells were incubated in a cell culture medium with Hoechst 33342 (Thermo Fisher Scientific) at a final concentration 5 μg/ml at 37 °C for 30 min prior analysis. Images were taken with Axio Observer.Z1 microscope equipped with AxioCam MRm camera, filter set excitation 450–490, emission 515–565 (GFP) and filter set excitation 335–383, emission 420–470 (Hoechst 33342), A-Plan ×10/0.25 Ph1 objective (Zeiss), and AxioVs40 module. A number of nuclei and inclusions were calculated with ImageJ 1.51j8 and 3d object counter plugin[55].

**Quantification of CGG containing mRNAs in COS7 cells**. For assessment of the level of mRNAs containing CGG repeats, transfection of COS7 with 100×CGG construct and ASOs and total mRNA isolation was performed as described in "miRNA level quantification". Next, cDNA was synthesized with the use of GoScript™ Reverse Transcriptase System (Promega) according to the manufacturer's instructions with the exception of the first-strand synthesis reaction temperature which was 37 °C. We used an anchored oligo(dT) primer containing oligo dT tract which allowed for exclusive reverse transcription of polyA+ RNAs. qPCR was performed with the use of iTaq™ Universal SYBR® Green Supermix (Bio-Rad) according to manufacturer's instructions and analyzed with the use of QuantStudio 7 Flex Real-Time PCR System machine (Thermo Fisher Scientific). To ensure amplification of 100×CGG construct mRNA but not DNA following primers were used: forward primer complementary to the 3′ part of GFP sequence and universal reverse (UR) primer. Primers for amplification of endogenous mRNA of FMR1 and NUB1 were set in 3′UTR of transcripts. Ct values were normalized against GAPDH and amplified with gene-specific forward primer set in 3′ part of mRNA and UR primer. All primers are listed in Supplementary Table S2. Fold differences in expression level were calculated according to the $2^{-\Delta\Delta Ct}$ method.

**Sucrose-gradient fractionation**. For assessment of mRNAs association with ribosomes, we used COS7 cells at 30–40% confluency, T75 bottles (75 cm$^2$), 20 μg of 100xCGG construct per bottle, and 9-nucleotide-long ASOs at 200 nM concentration. ~3.5–4 mln of cells were used to prepare the single sample. After 48 h, cells were incubated for 30 min in cell medium with cycloheximide (Merck) at a final concentration 100 μg/ml. Cells were washed with PBS, trypsinized, centrifuged, suspended in lysis buffer (20 mM Tris-HCl pH 7.5, 10 mM NaCl, 3 mM MgCl$_2$, 1 mM DTT, 0.3% Triton X-100, 50 mM sucrose, 1 mM RNasin (Promega), 100 μg/ml cycloheximide), and vortexed. Lysates were centrifuged at 10,000 × g for 10 min. Supernatants were transferred to new probes and proper volumes of 1 M NaCl and 1 M MgCl$_2$ were added to adjust final concentrations to 170 mM and 13 mM, respectively. Linear 15–45% sucrose gradient was prepared in a buffer (25 mM Tris-HCl pH 7.5, 25 mM NaCl, 5 mM MgCl$_2$, 100 μg/ml cycloheximide) in Open-Top Polyclear Tubes (tube size: 9/16" X 3 1/2", Seton Scientific) with the use of Biocomp Gradient Station (BioComp Instruments) and incubated for 30 min at

4 °C. Lysates were carefully loaded on gradients and ultracentrifuged at 160,000×*g* at 4 °C for 2 h 40 min in SW41 Ti swing-out rotor and Optima L100XPN Ultracentrifuge (Beckman Coulter). Next, 12 fractions of 0.97 ml were collected with the use of Piston Gradient Fractionator (BioComp Instruments) and 2110 Fraction Collector (Biorad) at a 0.3 mm/s speed, with a distance of 6.75 mm per fraction. Absorbance was monitored at 254 nm (A254) with an EM-1 Econo UV monitor (Biorad). Fractions were frozen at −80 °C and then used for total RNA isolation. 0.5 μl of GlycoBlue™ Coprecipitant (15 mg/ml, Thermo Fisher Scientific) was added to fractions. 5 M NaCl was added to obtain a final concentration of 0.1 M. Equal volume of isopropanol (~1 ml) was added, mixed, and incubated for 30 min at −20 °C. Samples were centrifuged at 12,000 × *g* for 10 min at 4 °C. Pellet was resuspended in 50 μl of ultrapure water and vortexed with 0.5 ml of TriReagent (Merck). Next, the total RNA was isolated according to the manufacturer's instructions with the addition of 1 μl of GlycoBlue™ Coprecipitant (15 mg/ml, Thermo Fisher Scientific) and sodium acetate 3 M pH 5.5 (Thermo Fisher Scientific) to a final concentration of 0.3 M to the aqueous phase. RNA was diluted in 15 μl of ultrapure water. In total, 5.6 μl of RNA solution was used for cDNA synthesis which was performed as described under "Quantification of CGG containing mRNAs in COS7 cells". PCR and gel electrophoresis were performed as described under "Alternative splicing analysis", with primers listed in Supplementary Table S2. qPCR was performed as described under "Quantification of CGG containing mRNAs in COS7 cells".

**Quantification of *FMR1* mRNA and pre-mRNA upon RNase H1 insufficiency**. For quantification of the effect of RNase H1 insufficiency and ASOs on *FMR1* mRNA and pre-mRNA level, fibroblasts were seeded on a 12-well plate and transfected at ~80% of confluency with siRNAs (future synthesis) at a final concentration of 15 nM. siRNA sequences are listed in Supplementary Table S2. After 24 h, the 11-nucleotide long (CGG$^{norm}$/- (2), CGG$^{norm}$/CGG$^{exp}$, CGG$^{exp}$/CGG$^{exp}$) or 9-nucleotide long (CGG$^{norm}$/- (1)) ASOs at 200 nM final concentration were delivered. Fibroblasts were harvested 48 h post ASOs delivery. The isolation of RNA was performed with the Total RNA Zol-Out™ D kit (A&A Biotechnology). In total, 300–500 ng of the total RNA was used for reverse transcription with GoScript™Reverse Transcriptase (Promega) and random primers (Promega). qPCR was performed with the use of iTaq™ Universal SYBR® Green Supermix (Bio-Rad) according to the manufacturer's instructions and analyzed with the use of QuantStudio 7 Flex Real-Time PCR System machine (Thermo Fisher Scientific) with primers listed in Supplementary Table S2. Ct values were normalized against *GAPDH*. Fold differences in expression level were calculated according to the $2^{-\Delta\Delta Ct}$ method.

**In vitro transcription**. In vitro transcription reactions presented in the main figures were performed on the 100×CGG construct linearized by Avr II which digest the genetic construct at the end of the 5′UTR sequence. The construct contains T7 polymerase promoter upstream to 5′UTR of *FMR1*. Reactions were performed with ~500 ng of template DNA in 10 μl of a mixture containing 1× transcription buffer (Promega), 10 mM DTT, 10 U of RNasin (Promega), rNTPs (0.5 mM each), 4 U of T7 polymerase (Promega) and (if required) 2.5 μM 9-nucleotide-long ASO-ctrl, ASO–CCG or ASO–CCG-Cy3. In the case of RNase H treatment, 1 U of RNase H (Thermo Fisher Scientific) was added to the mixture of proper samples at the beginning of the reaction. Samples without RNase H were treated with an equal volume of 50% glycerol to provide the same density of the mixture. The reactions were performed at 37 °C for either 20 min (the full saturation of transcript by ASO during the whole reaction) or 90 min (the full saturation of transcript only at the beginning of the reaction). Transcription reactions were stopped by the addition of 6×BLUE DNA Loading buffer (Blirt) which contains 60 mM EDTA. All in vitro transcription reaction products were analyzed on 1% agarose gels with or without ethidium bromide (0.5 μg/ml) run in 1 x Tris-Borate-EDTA buffer at 70 V for 2 h. To visualize the nucleic acid products, the agarose gels were scanned using Amersham Typhoon RGB Biomolecular Imager. Nucleic acids with intercalated ethidium bromide and the signal coming from ASO–CCG-Cy3 were detected using Cy3 filter. Intensities of the transcript signals were measured and quantitated with Multi Gauge 3.0 software (Fujifilm) and ImageQuant TL 8.1.0.0 (Cytiva). Each signal (gels stained with EtBr) was normalized upon the intensity of the signal coming from a genetic construct in the same lane. This procedure was applied to compensate for random variations in the sample signal intensities due to gel loading errors. Note that the first R-loop visualization was tested by two approaches: (1) In vitro transcription performed on the non-digested 100xCGG construct (Supplementary Fig. S3b, left panel), and (2) In vitro transcription performed on the 100xCGG construct digested with Avr II (Supplementary Fig.3b, right panel). In vitro transcription confirming R-loops formation was performed on the digested 100xCGG construct (Supplementary Fig. S3c). Detailed procedures are provided in the Supplementary Figures' description.

**Cytoplasm/nucleus fractionation**. For cell fractionation of the nuclei and cytoplasm, fibroblasts were seeded on a 100-mm plate and transfected at ~80% of confluency with 9-nucleotide-long ASOs at 200 nM final concentration. Fibroblasts were harvested 48 h post ASOs delivery, washed in ice-cold 1× PBS and centrifuged at 500 × *g* for 5 min. Cell pellets were resuspended by gentle pipetting in 1 ml

ice-cold HLB buffer (hypotonic lysis buffer: 10 mM Tris-HCl (pH 7.5), 10 mM NaCl, 3 mM MgCl₂, 0.3% (v/v) NP-40, 10% (v/v) glycerol), complemented with RNasin (Promega). Mix was incubated on ice 30 min and vortexed briefly. Cells were centrifuged at 2000 × *g* for 5 min at 4 °C. The supernatant (cytoplasmic fraction) was transferred to a new tube, kept on ice, and 5 M NaCl solution was added to adjust the NaCl concentration to 140 mM. The nuclei pellet was washed 4x by adding HLB, pipetting, and centrifuging at 500 × *g* for 2 min at 4 °C. The total RNA was isolated with the use of TRI Reagent (Merck) according to the manufacturer's instructions. cDNA synthesis and qPCR were performed as described under "Quantification of *FMR1* mRNA and pre-mRNA upon RNase H1 insufficiency" with primers listed in Supplementary Table S2.

**Western blot**. For analysis of proteins level CGG$^{norm}$/- (2), CGG$^{exp}$/CGG$^{exp}$ fibroblasts were seeded on a 100-mm plate and transfected at ~80% of confluency with 11-nucleotide long ASOs at 200 nM final concentration. Fibroblasts were harvested 48 h post ASO delivery. Cell extracts were prepared in lysis buffer (pH 8, 150 mM NaCl, 1% NP-40, 1× protease inhibitor (Roche) and 1 mM PMSF), vortexed, and frozen at −80 °C. Protein extracts were centrifuged at 11,300 × *g* for 10 min at 4 °C, measured by Pierce™ BCA Protein Assay Kit (Thermo Fisher Scientific), and heat-denatured for 5 min at 95 °C with the addition of standard sample buffer. Electrophoresis and wet transfer were performed with the use of Mini-PROTEAN Tetra System (Bio-Rad). In all, 30–40 μg of protein per well was separated in 10% SDS polyacrylamide gel in Laemmli buffer. Proteins were transferred to the nitrocellulose membrane (GE Healthcare) for 1 h, 100 V in Laemmli buffer with 20% methanol. Membranes were blocked in 5% nonfat dry milk in PBS with 0.1% Tween 20 (PBS-T) overnight at 4 °C. Incubation with antibodies was performed in following conditions: rabbit anti-FMRP antibody (ab17722, Abcam) 1:1000, rabbit anti-KANSL1L antibody (NBP2-14139, Novus Biologicals) 1:125, rabbit anti-DAZAP1 (PA5-41887, Thermo Fisher Scientific) 1:2000 in 5% nonfat dry milk in PBS-T for 4.5 h at 4 °C, mouse anti-GAPDH antibody (sc-47724, Santa Cruz Biotechnology) 1:8000/1:10,000 for 1 h at RT. Membranes were washed in PBS-T and incubated with horseradish peroxidase-conjugated secondary antibodies, anti-rabbit (AS09 602, Agrisera) 1:2500 in PBS-T for 1 h at RT or anti-mouse (sc-2005, Santa Cruz Biotechnology) 1:20,000 (anti-GAPDH) in PBS-T for 45 min at RT. Membranes were washed in PBS-T and covered with SuperSignal™ West Femto Maximum Sensitivity Substrate (Thermo Fisher Scientific) according to the manufacturer's instructions. Images were captured with the use of G:Box Chemi-XR5 (Syngene) and analyzed with the use of GeneTools 4.02 (Syngene). For analysis of proteins level CGG$^{norm}$/- (1), CGG$^{norm}$/- (3), CGG$^{norm}$/- (4), CGG$^{exp}$/- fibroblasts were seeded on 100 mm plate or 6-well plates and transfected at ~80% of confluency with 9-nucleotide long ASOs at 200 nM final concentration. Fibroblasts were harvested 48 h post ASO delivery. Cell extracts were prepared in lysis buffer (pH 8, 150 mM NaCl, 1% NP-40, 1× protease inhibitor (Roche) and 1 mM PMSF), vortexed, and frozen at −80 °C. Next, protein cellular extracts were centrifuged at 11,300 × *g* for 10 min at 4 °C, measured by Pierce™ BCA Protein Assay Kit (Thermo Fisher Scientific), and heat-denatured for 10 min at 70 °C with the addition of Bolt LDS buffer (Invitrogen). In total, 20–30 μg of protein per well was separated in Bolt™ 4–12% Bis-Tris Plus gel (Invitrogen) in Bolt™ MES SDS Running Buffer. Proteins were transferred to PVDF transfer membrane (GE Healthcare) for 1 h, 100 V in Leammli buffer with 20% methanol. Western blot experiments were performed using SNAP ID Protein Detection System (Merck Millipore). Membranes were blocked with 0.125% nonfat dry milk in TBS with 0.1% Tween 20 (TBS-T) for 20 min. Incubation with antibodies was performed in following conditions: rabbit anti-FMRP antibody (ab17722, Abcam) 1:500 in 0.125% nonfat dry milk in TBS-T for 1 h 10 min, mouse anti-GAPDH antibody (sc-47724, Santa Cruz Biotechnology) 1:10,000 for 15 min, rabbit anti-SLC40A1 antibody (NBP1-21502, Novus Biologicals) 1: 1000 for 1 h, rabbit anti-QKI antibody (ab126742, Abcam) 1:1000 for 1 h, rabbit anti-VKORC1L1 antibody (PA5-48618, Thermo Fisher) 1:300 for 1 h, rabbit anti-alpha-tubulin antibody (ab52866, Abcam) 1:10,000 for 15 min. Membranes were washed in TBS-T and incubated with horseradish peroxidase-conjugated secondary antibodies–anti-rabbit (A9169, Merck) 1:20,000 for 15 min or anti-mouse (A9044, Merck) 1:20,000 and washed with TBS. Antibody–antigen complexes were visualized by enhanced chemiluminescence (ECL) using Luminata Forte HRP Substrate (Merck) and detected with G:BoxSystem (Syngene). For analysis of protein level in mice hippocampus tissues, samples were lysed with the help of a dounce homogenizer (Wheaton) in denaturing lysis buffer (1% sodium dodecyl sulfate, 1 mM EDTA, 20 mM Tris-HCl, 1 tablet Pierce protease inhibitor (Thermo Fisher Scientific)). Samples were homogenized by agitation for 5 min at 95 °C. After centrifugation for 1 min at 16,000 × *g*, the supernatant was collected for further processing. The protein concentration of each sample was determined using the Bradford assay (Bio-Rad) according to the manufacturer's instructions. In all, 20 μg protein from each sample was separated by SDS-PAGE (12%) and transferred to PVDF membranes (Merck). Membranes were first blocked in Intercept Blocking Buffer (Li-COR), then incubated overnight at 4 °C with primary antibodies: rabbit anti-GFAP (NB300-141, Novus Biologicals) 1:5000, rabbit anti-TNF-alpha (ab66579, Abcam) 1:750, rabbit anti-CD68 (ab125212, Abcam) 1:500, mouse anti-alpha-tubulin (T6199, Merck) 1:10,000 and antigen–antibody complexes were visualized via incubation with fluorescent secondary antibodies: anti-rabbit IRDye 800CW (926-32211, Li-COR) 1:15000 and anti-mouse IRDye 680RD

(926-68070, Li-COR) 1:15000 for 1 h at RT. Florescent blotted membranes were scanned with the Odyssey scanner (LI-COR) and the signal associated with the antigen/antibody complexes were analyzed using FIJI-ImageJ software. For analysis of protein level in mice cortex tissues, samples were placed in tubes with 50 µl of lysis buffer (pH 8, 150 mM NaCl, 1% NP-40, 1× protease inhibitor (Roche), and 1 mM PMSF), homogenized with the use of Omni Tissue Homogenizer (Omni International), incubated for 30 min on ice, vortexed and frozen at −80 °C overnight. Then samples were processed as samples derived from CGG[norm]/- (1), CGG[norm]/- (3), CGG[norm]/- (4), CGG[exp]/- fibroblasts. In total, 20 µg of proteins were separated. For detection of FMRP, QKI, VKORC1L1, and SLC40A1 procedures were the same as for fibroblasts samples. For detection of AIF1 and PPP1R1B, membranes were blocked with 1% bovine serum albumin (BSA) in TBS-T for 20 min. Incubation with antibodies was performed in following conditions: rabbit anti-alpha tubulin antibody (ab52866, Abcam) 1:10,000 in 1% BSA in TBS-T for 15 min, rabbit anti-AIF1 antibody (019-19741, Wako) 1:500 for 1 h, rat anti-PPP1R1B antibody (MAB4230, R&D Systems) 1:500 for 1 h. Membranes were washed in TBS-T with 0,5 M NaCl and incubated with horseradish peroxidase-conjugated secondary antibody anti-rabbit (A9169, Merck) 1:20,000 for 15 min or rhodamine-conjugated (TRITC) anti-rat (712-025-153, JIR) 1:10,000 for 15 min. Membranes were washed with TBS with 0.5 M NaCl and antibody–antigen complexes were visualized Luminata Forte HRP Substrate (Merck) and G:BoxSystem (Syngene) for anti-rabbit and by Amersham Typhoon RGB Biomolecular Imager with Cy5 filter for anti-rat antibody. Intensities of the protein signals were measured and quantitated with ImageQuant TL 8.1.0.0 (Cytiva). For detection of the signal from different antibodies, the membrane was cropped or washed with stripping buffer (1.5% glycine, 0.1% SDS, 1% Tween 20, pH 2.2).

**Animals**. All experimental procedures were approved by local ethics committee Landesverwaltungsamt Sachsen-Anhalt (CEEA# 42502-2-1219UniMD) and met the guidelines of local and European regulations (European Union directive no. 2010/63/EU). Inducible bitransgenic animals (referred to as P90CGG throughout the text) were obtained by crossing single transgenic mutants carrying a 90CGG repeat tract along with a tetracycline responsive element (TRE-90CGG) with the transgenic prion protein-reverse tetracycline transactivator (PrP-rtTA) driver line, as described previously[32]. Mice were weaned at 4 weeks of age and group-housed in standard laboratory cages containing standard bedding material and Nestlets™ nesting material (Ancare). Animals were kept in temperature and humidity-controlled conditions with an inverted 12-h light/12-h dark cycle. Water and food were provided ad libitum. Genotyping was performed from tail biopsies that were incubated overnight at 55 °C in DirectPCR tail lysis reagent (Viagen) and 0.3 mg/ml proteinase K (Carl Roth). Following heat inactivation, 1 µl from each sample was used for the genotyping PCR, performed according to the previously described protocol[56].

**Doxycycline (DOX) administration**. P90CGG mice were provided 4 g/kg DOX-containing food pellets (ssniff) for a period of 12 weeks starting from the day of weaning. At the end of the 12-week of exposure, DOX-containing food pellets were replaced with standard mouse feed until sacrifice. The same DOX induction schedule with P90CGG mice has previously shown to result in motor deficits and a high load of intranuclear inclusions in the cerebellum[33].

**Stereotactic surgery and ASO administration**. At weaning, P90CGG were randomly allocated to one of the two treatment groups (saline versus ASO–CCG) with animals from the same litter being represented in both treatment groups. 8-week old mice were anesthetized in an isoflurane anesthesia system (Rothacher) with 5% isoflurane in $O_2/N_2O$ mixture and placed on a stereotactic frame (World Precision Instruments). Anesthesia was maintained throughout the surgery at 1.5–2.0%. A 28 G, brain infusion cannula (Plastics1) with 2.5-mm cut length that is attached via a catheter to a primed osmotic pump (Alzet model: 1004; Durect) filled with 100 µl of 11-nucleotide-long ASO solution (~5.88 µg/µl) was inserted at coordinates M/L: −0.11 (directed laterally to the right), A/P: −0.05 relative to bregma resulting in an intracerebroventricular infusion at the right lateral ventricle. The cannula was secured to its location with Paladur® dental cement (Kulzer). Pumps were subcutaneously implanted on the back of the mice posterior to the scapulae. Osmotic pumps were removed under anesthesia at the end of the 4-week infusion schedule. Control animals were infused with 0.9% saline solution. Cannula placement was visually validated for each animal during cryosectioning.

**Motor behavior**. At the end of the 12-week DOX induction period, P90CGG mice were subjected to a 3-day motor performance test on Rotarod equipment (Ugo Basile). On day one, mice were trained at 15 rpm constant speed on the rotating rod for a maximum of 60 s in four trials. On the second day, animals were tested for each of the five different speeds (15, 24, 31, 36, 40 rpm) in two trials for a maximum of 60 s. The third day consisted of testing the mice during a 5-min accelerated ramp from 4 to 40 rpm in four trials. In each trial, the latency to fall off the rod was registered.

**Tissue processing**. Animals were anesthetized in an isoflurane chamber, sacrificed by decapitation, and then brains were immediately removed. Right hemispheres are drop fixed overnight at 4 °C in 4% paraformaldehyde/PBS and cryoprotected in

30% sucrose/PBS containing 0.02% sodium azide. Fixed brain samples were then embedded in Tissue-Tek® O.C.T.™ compound (Sakura) and frozen in liquid nitrogen-cooled isopentane. Frozen blocks were cut in 7 µm thick sagittal sections in a freezing microtome (Leica). Sections were mounted on poly-L-lysine coated slides, air-dried at room temperature, and stored at 4 °C for fluorescent immunohistochemistry. For RNA-seq, the striatum and the cortex (excluding the prefrontal cortex) of the mice have been dissected out from the left hemispheres in ice-cold ACSF and immediately snap-frozen in liquid nitrogen. The samples were kept at −80 °C until further processing.

**FMRpolyG immunohistochemistry**. Slide mounted sections were treated with 0.01 M sodium citrate (pH = 6.0) in a microwave oven for antigen retrieval. An extra antigen retrieval step with proteinase K. (5 µg/ml) was included to better visualize the inclusions. Sections were first blocked for endogenous biotin and then blocked for mouse Ig using M.O.M. reagents (Vectorlabs). Sections were incubated overnight at 4 °C in mouse-FMRpolyG (8FM[57]; 1:200) antibody. Following biotinylated secondary antibody incubation (mouse on mouse immunodetection kit, BMK-2202, Vector Labs), antigen–antibody complexes were visualized by incubation in Cy5-tagged streptavidin (Invitrogen). Sections were counterstained with DAPI and coverslipped with Immu-Mount™ (Shandon). The granular layer of the cerebellum lobule X was selected for this quantification. Fluorescent photomicrographs of cerebellum lobule X with a z-step size of 2 µm were registered under an epifluorescence microscope (Leica) at 630x magnification and analyzed with open-source image processing software Fiji 2.0.0-rc-69/1.52p. 400 DAPI + nuclei that lie within the granular layer of the lobule X were counted per mice by a researcher who was blind to the treatment groups. For this quantification, random images of 30 µm by 30 µm generated from whole photomicrographs of lobule X by a custom cell counter script developed for Fiji software were used. For each DAPI + nuclei that were counted, the script recorded through the researcher's input whether or not it contains a FMRpolyG+ intranuclear inclusion and its longitudinal size. From the data of the identified inclusions, a percentage and an average size value were calculated for each mouse. Statistics have been performed from these values.

**RNA-seq and bioinformatics analysis**. Frozen cortex or striatum samples were placed in tubes with 600 µl of TriReagent (Merck) and homogenized with the use of Omni Tissue Homogenizer (Omni International). Total RNA was isolated with the use of Direct-zol™ RNA MiniPrep kit (Zymo Research) according to the manufacturer's instructions with DNase I treatment. RNA quantity was checked with the use of Qubit® RNA BR Assay Kit and Qubit Fluorometer (Thermo Fisher Scientific) according to manufacturer's instructions, RNA quality was checked with the use of RNA 6000 Nano Kit and 2100 Bioanalyzer System (Agilent) according to the manufacturer's instructions. Library preparation and total RNA sequencing (2×100 bp) were performed by CeGaT (Germany) with the use of 100 ng of RNA, TruSeq Stranded Total RNA kit, and NovaSeq 6000. Demultiplexing of the sequencing reads was performed with Illumina bcl2fastq (2.19). Adapters were trimmed with Skewer (version 0.2.2)[58]. Reference *Mus musculus* (GRCm38, primary assembly) genome and annotations were downloaded from Ensembl (ver. 91 or 93). Quality and adapter trimming of short reads was performed using Trimmomatic 0.39[59]. Short reads matching known rRNA sequences were removed using HISAT2 2.1.0 aligner[60]. Read quality reports before and after quality filtering were prepared using FastQC 0.11.5 software (http://www.bioinformatics.babraham.ac.uk/projects/fastqc/). Filtered reads were aligned to the reference genomes using STAR 2.7.1a algorithm[61]. Read mapping reports were created using Qualimap 2.2.2 software[62]. RSEM 1.3.0 (RNA-Seq by Expectation Maximization[63]) was used to quantify the expression values of genes. Hierarchical clustering of RNA-seq samples (Pearson correlation metric, centroid linkage) based on the expression values of all genes was performed using standard R 3.3.1 functions (R Core Team 2016) and variance stabilizing transformation provided by DESeq2 package[64]. Differential expression analysis between designated groups of samples was performed using voom+limma[65] pipeline with settings described in[66]. The edgeR 3.14.0 package was used for initial data filtering and normalization using the TMM method[66]. Voom function from the limma 3.28.21 package[66] was employed to estimate the mean-variance relationship in the data. Afterward, linear modeling and empirical Bayes moderation were applied to identify differentially expressed genes between groups of interest as described in ref. [66]. P value was generated using moderated *t*-statistic and adjusted for multiple testing using Benjamini–Hochberg's method (adjusted *p*-value, adj.P.Val). adj.P.Val threshold of 0.01 and fold change threshold of 1.1 were used during the analysis. RNA-seq data were stored in NCBI SRA database. SRA accession: PRJNA577423.

**Gene ontology analysis**. For gene ontology analysis, data from differential expression analysis of striatum or cortex RNA-seq results were used. List of genes expressed on arbitrarily determined level (AveExpr > 1; AveExpr –average expression across all samples, in log2 CPM) and significantly upregulated or downregulated (Adj.P.Val ≤ 0.05; Adj.P.Val – Benjamini–Hochberg false discovery rate adjusted P value) was extracted for saline vs control and ASO–CCG vs control groups. A reference list of genes was prepared from all genes detected in striatum or cortex tissue which were expressed on an arbitrarily determined level (AveExpr > 1).

The analysis was performed with the use of PANTHER14.1 program. The analysis type was PANTHER Overrepresentation Test, annotation version and the release date was GO Ontology database released 2019-02-02. Annotation datasets were GO biological process complete, GO molecular function complete, and Reactome pathways. Test type was Fisher's exact with the use of Bonferroni correction for multiple testing.

**FL-SDS-PAGE.** For assessment of soluble FMRpolyG-GFP after ASO–CCG treatment, we used COS7 cells, 48-well plate, 100 ng of 100×CGG construct, 100 ng of 100×CGG-mCherry construct, and 50 ng of reference mCherry construct per well and 11-nucleotide-long ASOs at 125 or 200 nM. After 48 h cells were harvested and proceeded according to FL-SDS-PAGE protocol described with details in ref. [23]. Shortly, cells were lysed, vortexed, and sonicated. In total, 10 µl of lysate without previous heat denaturation was electrophoretically separated in 12% polyacrylamide SDS gel. The fluorescence signal was captured directly in the gel with the use of Amersham Typhoon RGB Biomolecular Imager.

**Quantification of mRNAs and pre-mRNAs containing CGG and CCG repeats.** For analysis of mRNAs and pre-mRNAs level fibroblasts were seeded on 100-mm plate and transfected at ~80% of confluency with 11-nucleotide-long (CGG$^{norm}$/-(2), CGG$^{norm}$/CGG$^{exp}$, CGG$^{exp}$/CGG$^{exp}$) or 9-nucleotide long (CGG$^{norm}$/- (1), CGG$^{exp}$/-) ASOs at 200 nM final concentration. Fibroblasts were harvested 48 h post ASOs delivery. RNA isolation, cDNA synthesis and qPCR were performed as described under "Cytoplasm/nucleus fractionation" with primers listed in Supplementary Table S2. PCR and gel electrophoresis were performed as described under "Alternative splicing analysis" with primers listed in Supplementary Table S2.

**Assessment of efficiency of ASOs delivery via gymnosis using flow cytometry.** Assessment of efficiency of ASOs delivery via gymnosis was performed as described under "Flow cytometry". 9-nucleotide-long ASOs (ASO–CCG and ASO–ctrl) were delivered via transfection at 200 nM concentration or directly to the culture medium (gymnosis) at 200 or 400 nM concentration (ASO–CCG).

**Quantification of the expression level of TRE-nCGG-eGFP transgene and immune system-related markers in P90CGG mouse.** The expression levels of TRE-nCGG-eGFP transgene and immune system-related markers upon ASO–CCG delivery were quantified from total RNA isolated from striatal and cortical tissue, respectively, of saline or ASO–CCG treated mice. cDNA synthesis and qPCR were performed as described under "Quantification of *FMR1* mRNA and pre-mRNA upon RNase H1 insufficiency" with primers listed in Supplementary Table S2.

**Heatmap.** Heatmap was created based on differential expression analysis results from the striatum of control and P90CGG mouse. Only genes with Adj.P.Val<0.05 were selected for further analysis. The set contained 417 genes from the saline vs control group. In the final results, one gene "ENSMUSG00000050121_Opalin" was removed because of its outlier characteristics. Samples marked with numbers in the range 1–4 belonged to the saline group, samples marked with numbers in the range 5-8 belonged to the ASO–CCG group, and samples marked with numbers in the range 9–12 belonged to the control group. Genes present in twelve samples were visualized using a three-color heatmap. Data for all samples were organized in order from left to right in 12 columns plus the first column describing a particular gene. For every gene (row) and every sample in a given gene, a Z-score was calculated using the average and standard deviation of four control samples. Resulting Z-scores for every sample in every row (gene) were saved in tab-delimited text files with a header and then converted into the heatmap. Heatmap was generated using R "pheatmap" library version 1.0.12. Both rows (genes) and samples (columns) were clustered using complete-linkage method (clustering_method = "complete" option in pheatmap package). Z-scores for every gene (rows) and samples (columns) are represented with 50 shades between orange, white, and green. Results are centered so that Z-score equal to zero is represented by white.

**Statistics and reproducibility.** All data obtained in this study were processed and analyzed with the use of Microsoft Excel. Statistical analysis of all experiments performed on cell lines as well as western blots performed on mice tissues was done using unpaired two-tailed Student's *t* test. The normality of the data has been verified using Lilliefors test. Tests that result in $P < 0.05$ have been reported to be statistically significant. The symbols; *,**,*** represent values of $P < 0.05$, $P < 0.01$, $P < 0.001$, respectively. All data were analyzed using Statistica software version 10 (TIBCO Software) and Prism software version 7 (GraphPad). Error bars represent standard deviation (SD). Statistical analysis on the behavior data was done using two-way repeated-measures ANOVA for training and constant speed rotarod, and unpaired two-tailed Student's *t* test for accelerating rotarod test. The FMRpolyG+ nuclei counts and size were analyzed using unpaired two-tailed Student's *t* test. The normality of the data has been verified using Shapiro–Wilk test whenever *t*-statistics have been used to demonstrate statistical significance. Tests that result in

$P < 0.05$ have been reported to be statistically significant. The symbols; *,**,*** represent values of $P < 0.05$, $P < 0.01$, $P < 0.001$, respectively. All data were analyzed using Prism software version 7 (GraphPad). Error bars represent SEM for rotarod tests and SD for the FMRpolyG+ nuclei counts and size. Pearson correlation and Mann–Whitney test applied to RNA-seq data analysis were calculated using Prism software version 7 (GraphPad). All cellular and in vitro experiments presented in this work was repeated at least two times with similar results.

**Reporting summary.** Further information on research design is available in the Nature Research Reporting Summary linked to this article.

## Data availability

RNA-seq data generated and analyzed during this study are available in the NCBI SRA database. SRA accession: PRJNA577423. Publicly available datasets from https://www.ensembl.org/ were used in this study. The code used to generate the heatmap presented in Supplementary Fig. S5 is available in the GitHub repository, https://github.com/MagdalenaDerbis/R-script-to-generate-heatmap/blob/main/Supplementary%20R%20script%20to%20generate%20heatmap.R. Other data that support the findings of this study are available from the corresponding author upon reasonable request. Source data are provided with this paper.

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

## Acknowledgements

The authors wish to thank Nicolas Charlet-Berguerand for providing 8FM; anti-FMRpolyG antibody and 20xCGG, 60xCGG, and 100xCGG genetic constructs, Anita Bhattacharyya for providing C0603, FX11-02, and WC26 fibroblasts and Paul J. Hagerman for providing C6 and F3 fibroblasts. This work was supported by the National Centre for Research and Development grant ERA-NET-E-Rare- 2/III/DRUG_FX-SPREMUT/01/2016 (to K.S.), by the Ministry of Science and Higher Education of the Republic of Poland, from the quality-promoting subsidy, under the Leading National Research Centre (KNOW) program for the years 2012-2017 [KNOW RNA Research Centre in Poznan (No. 01/KNOW2/2014)], by the Foundation for Polish Science, TEAM POIR.04.04.00-00-5C0C/17-00 (to K.S.), by the Initiative of Excellence–Research University (05/IDUB/2019/94) at Adam Mickiewicz University, Poznan, Poland, and from the Poznan RNA Research Centre (to K.S.), by ERA-Net for Research Programmes on Rare Diseases grant 01GM1505 "Drug_FXSPreMut" (E-Rare-2 JTC 2014; to O.S.) and by the Deutsche Forschungsgemeinschaft (DFG, German Research Foundation) – 362321501/RTG 2413 SynAGE to O.S.

## Author contributions

K.S., O.S., and R.H. designed the study. M.D., E.K., D.N., M.S., A.P., and K.T. performed experiments and analyzed the data. E.K. performed all experiments on mouse models (Fig. 4c, d, e and Supplementary Fig. S4b), staining of FMRpolyG (Fig. 4f and Supplementary Fig. S4c), and western blots on hippocampus mouse tissues (Fig. 5e and Supplementary Figs. S6e, f). D.N. tested the effect of RNase H and ASOs on FMR1 transcript in fibroblast and in vitro (Fig. 3a–c and Supplementary Figs. S3a–c), assessed efficiency of ASOs delivery (Supplementary Fig. 4a), assessed level of CGG containing mRNAs (Supplementary Fig. S6c), and performed western blots on cortex mouse tissues and fibroblast cell lines (Figs. 3e, 5c and e and Supplementary Fig. S6d, e). M.S. performed cellular fractionation (Fig. 3d and Supplementary Fig. 3e), assessed level of FMR1 transcript and other CGG containing mRNAs (Supplementary Figs. S3d and S6c),

quantified FMRP after ASO treatment (Fig. 3e), and level of mRNAs of immune system-related toxicity markers in mouse cortex (Fig. 5d). A.P. quantified level of miRNAs (Fig. 1e and Supplementary Figs. S1c and d), assessed viability and apoptosis of cells after ASO treatment (Fig. 1c, d), and prepared Supplementary Table S1. K.T. assessed ASO affinity to CGG repeats in vitro (Fig. 1b and Supplementary Fig. S1a). K.S. performed the analysis of gene expression changes in RNA-seq data (Fig. 5a, b and Supplementary Fig. S6b). M.D. performed all other experiments and prepared figures. M.D., E.K., and K.S. wrote the paper.

## Competing interests
The authors declare no competing interests.
