## [Peer Review File · Nature Communications]

Reviewers' Comments:

Reviewer #1:

Remarks to the Author:

These authors have reported remarkable research regarding treatment of FXTAS first at a cellular level and then in vivo with the transgenic FXTAS mouse model. The use of short antisense oligonucleotide steroid blockers (ASOs) for CCG (ASO-CCG) had beneficial effects on removing R-loops, releasing sequestered proteins such as DRISHA and Sam-68 with effects on the miRNA levels and in reducing the FMRpolyG protein levels and inclusion formation. The use of the FXTAS mouse model showing the effects of the infusion of ASO-CCG into the ventricles with improvements in the motor scores on the rotorods and with reduction of the inclusions and FMRpolyG levels was wonderful. Finally the rescue by ASO-CCG of the transcriptome of many genes dysregulated by FXTAS was remarkable, however the up-regulation of many immune regulated genes by this treatment is ominous. Also the lowering of FMRP levels in those with FXTAS is also concerning for the clinical use of this treatment. However, this paper is a tour de force for FXTAS treatment and I congratulate these authors.

Reviewer #2:

Remarks to the Author:

Summary

With this work, the authors demonstrate that an ASO can alleviate pathology in a cellular model of a FXTAS, a trinucleotide repeat disorder, and produce modest behavioral and cellular benefits in a mouse model of the same disorder. Their data suggest that ASOs might one day serve as useful therapeutic options for patients with both FXTAS and other repeat expansion diseases. However, for the community to truly appreciate the therapeutic potential of the ASOs explored by this paper, the authors need additional information regarding the toxicity of their ASOs in both cells and animals. Claims regarding toxicities and cellular mechanisms should be unambiguously supported by data.

Specific Concerns

- ASO toxicity is unclear in both cellular and animal experiments
 - o The authors acknowledge that their ASO might generate off-target toxicity by affecting non-FXTAS genes that harbor CGG repeats, but they stop short of clearly characterizing the severity of this issue. While off-target effects are a common problem among ASOs, this seems particularly relevant here since per their statement 1.5% of transcripts have these repeats and the ASOs will presumably bind to these. They briefly note that their ASO increases transcription of off-target genes (Fig. S2d, Fig. 4h) and possibly decreases their translation rate (Fig. S2e), but they also report that western blot quantification fails to show a change in protein levels of two of these proteins (Fig. 3d), ultimately concluding that "the translation from mRNAs containing CGGexp is reduced selectively." This conclusion is premature both because the failure of their protein quantifications to demonstrate a statistically significant change in protein levels upon ASO administration does not mean that a significant biological change does not occur (particularly in the absence of a power analysis to the contrary) and because of the conflicting ribosomal data from Fig. S2e.
 - o It would be helpful to show direct toxicity and/or survival assays in both cells and animals that could alleviate or substantiate these concerns, ideally including assays that separate sequence-specific toxicities from general ASO toxicities. It needn't be the case that the authors' ASO is completely free of toxic off-target effects, but if clinical application is the ultimate goal, some exploration of the magnitude of these toxicities is warranted.
- Title includes "reversal", which is a strong statement regarding the effect of the ASOs. This reviewer does not see evidence of reversal. Please provide this evidence. Overall, effects appear to be modest slowing of progression.
- Long-term effects on the mouse model have not been characterized
 - o The authors note that "an interesting question to be addressed is whether this intervention is

sufficient to provide long-term protection or merely slows down the pathogenic process.” The work would be improved by examining the long-term consequences of ASO treatment. The authors should comment on feasibility of this test in their system and why not completed here.

- Mouse model improvements are not given in context of control mouse performance
 - o Rotarod performance data from control mice in the authors’ colony are not provided in Fig. 4c through Fig. 4e, so it is difficult to assess the severity of disease and magnitude of improvement in the treated and untreated model mice. It would be helpful to describe the severity of disease in their model mice, including whether they die from their disease and the time course of their symptoms.
- Claims of nuclear retention are not clearly supported
 - o The authors cautiously claim that the nuclear mRNA changes in Fig. 3c might suggest increased nuclear retention, but given the small effect size in the nucleus and larger change in the cytoplasm and the cell overall, it seems more likely that the nuclear changes occurred as a product of overall transcriptional changes.
 - Claims of R-loop suppression as the mechanism for transcriptional changes are not clearly supported
 - o The authors suggest that the FMR1 transcriptional changes they observe occur due to R-loop suppression, but they fail to show clear evidence to support this. They show that the ASO’s effect on transcription appears RNase-dependent and they label an R-loop band in Fig. 3b, but clearly demonstrating that R-loop suppression specifically is the mechanism at play would require additional data and a more thorough explanation.
- Minor
 - o Please provide more details regarding “ASO-ctrl” and rationale behind it
 - o Make it clear which aso is used in which experiments
 - o Fig 1b: n? error bar meaning?
 - o R-loops label on fig 3b is not obvious; should be explained in legend or elsewhere if authors really want to call attention to it
 - ♣ Supposed to note reduction in R-loop band when ASO given?
 - o Fig 3 c-e: why no data on control mice?

We thank Reviewers for considering our manuscript. We appreciate your input, dedicated time, thoughtful remarks and constructive suggestions. We revised the manuscript according to reviewers' comments. We believe that the new version of the manuscript is substantially improved after performing the recommended experiments and corrections. We hope we can satisfy all the concerns and incorporate all the suggestions.

Answers for specific questions are specified bellow and changes in main text are marked in red.

Reviewer #1

“the up-regulation of many immune regulated genes by this treatment is ominous.”

Delivery of long and short nucleic acids and their analogs (including ASOs) to cells and tissues is known to induce immune response, what depends on chemistry, length and sequence of ASOs. Driven by reviewer's concern we analyzed expression of immune system-related and toxicity-related markers reported in the literature and in our RNA-seq data from ASO-CCG treated mice. The expression of a few markers was significantly elevated on RNA level (e.g. *Gfap* and *Cd68*) and expression of many others was unchanged (**new Supplementary Table S5**). Moreover, we checked protein levels of GFAP, a marker of gliosis (both total and proteolytic fragment considered as the sign of glial injury¹), TNF- α , a marker of inflammation, microglia markers, CD68 and AIF1, as well as PPP1R1B, a marker of medium spiny neurons and none of them show statistically significant changes in ASO-CCG-treated mice (**new Figure 5c**). However, we observed a trend towards an increase in the levels of GFAP proteolytic fragments in ASO-CCG treated mice. Therefore, we conclude that treatment with ASO-CCG indeed induces immune system in both tested brain regions (hippocampus and cortex) but induction is not severe, especially on a protein level.

In the new version of the manuscript we added new **Supplementary Table S5** and new **Figure 5**, as stated above, containing analysis of immune related toxicity markers on mRNA and protein levels in ASO-CCG treated mice. Results were described in a last subchapter. Moreover, we added new results concerning global cellular toxicity of ASO-CCG to **Figure 1** (**new Figures 1b and c**). They did not show any deleterious effect of short LNA-based ASOs used in our experiments.

“the lowering of FMRP levels in those with FXTAS is also concerning for the clinical use of this treatment.”

Following the Reviewer's concern we decided to check the effect of ASO-CCG on FMRP production in two additional fibroblast cell lines, one with normal CGG repeat number, and one carrying CGG^{exp} in *FMR1* gene. None of those cell lines showed significant reduction of FMRP. Overall, only one cell line carrying two alleles with CGG^{exp} out of six examined (four with normal CGG repeat number and one with CGG^{exp}) showed significant decrease of FMRP upon ASO-CCG treatment (**modified Figure 3e**). Moreover, we checked FMRP level in brain tissue of ASO-CCG treated mice and it remained unchanged in comparison to saline treated mice, although, murine *Fmr1* contains only 6 uninterrupted CGG repeats (**new Figure 5c**). Therefore, we conclude that steady-state level of FMRP is generally stable; however, could be significantly lowered in some cases. Therefore, we believe that reasonable solution is to establish proper dose of ASO-CCG for potential clinical use. The ideal dose should be a compromise between beneficial alleviation of FXTAS pathological changes at a molecular level and reduction of FMRP which does not cause any adverse effects. Defining of such dose and regime of treatment should be the direction of future studies.

In the new version of the manuscript we extended **Figure 3e** containing analysis of FMRP level in fibroblasts (we added two new cell lines and two cell lines presented previously as a separate Supplementary Figure S3d) and created new **Figure 5c** with analysis of murine FMRP. Moreover, we modified the description of the results to better reflect our new findings.

Reviewer #2

Specific Concerns

- **ASO toxicity is unclear in both cellular and animal experiments**

- **The authors acknowledge that their ASO might generate off-target toxicity by affecting non-FXTAS genes that harbor CGG repeats**, but they stop short of clearly characterizing the severity of this issue. While off-target effects are a common problem among ASOs, this seems particularly relevant here since per their statement 1.5% of transcripts have these repeats and the ASOs will presumably bind to these. They briefly note that their ASO increases transcription of off-target genes (Fig. S2d, Fig. 4h) and possibly decreases their translation rate (Fig. S2e), but they also report that western blot quantification fails to show a change in protein levels of two of these proteins (Fig. 3d), ultimately concluding that “the translation from mRNAs containing CGGexp is reduced selectively.” This conclusion is premature both because the failure of their protein quantifications to demonstrate a statistically significant change in protein levels upon ASO administration does not mean that a significant biological change does not occur (particularly in the absence of a power analysis to the contrary) and because of the conflicting ribosomal data from Fig. S2e.

Thank you for raising this issue. We fully agree with the reviewer that relatively high incidence of genes containing CGG repeats together with ASO-CCG capacity to globally change their expression oblige us to look more closely to this problem. Therefore, we performed additional experiments that lead to some substantial changes in the manuscript. Based on RNA-seq data from brain of FXTAS mouse model we chose three additional genes containing CGG tracts which expression on mRNA level was elevated after ASO-CCG administration, *QKI/Qk* containing 8 uninterrupted CGG repeats in human gene and 7 CGG repeats in murine gene, *SLC40A1/Slc40a1* containing 7 and 6 CGG repeats, respectively, and *VKORC1L1/Vkorc1l1* containing 2x6 and 8 CGG repeats, respectively. We checked the level of protein products of these genes in FXTAS and healthy fibroblasts and, more importantly, in mouse brain tissue. Western blot analysis showed slightly decreased level of all proteins in healthy fibroblasts (statistically significant only for *VKORC1L1* and *QKI*) but no changes in FXTAS fibroblasts and mouse cortex were observed (**new Figures 5c** and **new Supplementary Figures S6c** and **d**).

Also lowering of FMRP level could be considered as the off-target effect of ASO-CCG. Therefore, we checked FMRP level in two additional ASO-CCG treated fibroblast cell lines and in brain tissue of ASO-CCG treated mice. Only one cell line (with expanded CGG repeats) out of six tested cell lines showed a significant decrease of FMRP upon ASO-CCG treatment (modified **Figure 3e**). FMRP level in brain tissue of ASO-CCG treated mice remained unchanged in comparison to saline treated mice, although, murine *Fmr1* contains only 6 uninterrupted CGG repeats (**Figure 5c**). Overall, ASO-CCG influences, to a minor extent, the protein level of some off-target genes including *FMR1*. On the other hand, considering significant increase of some mRNAs containing short and long CGG repeats our results suggest that ASO-CCG reduces efficiency of translation of these mRNAs.

It was not our intention to conclude that “the translation from mRNAs containing CGGexp is reduced selectively”. We changed all fragments which could somehow suggest that including the title of subchapter “**ASO-CCG increases the transcription of *FMR1* locus and reduces efficiency of FMRP biosynthesis**”.

In the new version of the manuscript we added new **Figure 5** (mouse brain tissue) and **Supplementary Figure S6** (fibroblast cell lines) concerning ASO-CCG-related toxicity, including the off-target effect. Results were described in a new paragraph. Moreover, FMRP level in fibroblasts was presented on extended **Figure 3e** and described in the text.

o It would be helpful to show direct toxicity and/or survival assays in both cells and animals that could alleviate or substantiate these concerns, ideally including assays that separate sequence-specific toxicities from general ASO toxicities.

In the new version of the manuscript this concern is addressed more carefully. So far, our data and observations had not indicate clear general toxicity of ASO-CCG in both cellular and mouse models. For example, concluding from flow cytometry experiments, in population of cells transfected with control plasmid the percent of dead cells (positive for propidium iodide; PI) was similar in cells treated with either ASO-CCG, ASO-ctrl or treated only with transfection agent (see **Figure 1 below**). However, we agree with the reviewer, that toxicity can be more specific and should be investigated with more details. Thus, we conducted additional experiments regarding viability and induction of apoptosis upon ASO-CCG treatment in COS-7 cells. Time-course experiment again showed that the viability of cells treated with ASO-CCG, ASO-ctrl and transfection agent only is almost identical and significantly higher than treated with a positive control, ASO showing high toxicity (**new Figure 1c**). Moreover, ASOs did not elevate markers of early apoptosis (**new Figure 1d**).

To assess ASO-CCG toxicity in mice we searched literature towards markers of toxicity related to treatment of cells and tissues with short nucleic acids. Next, we checked expression of those markers in mouse brain in our RNA-seq results. Expression of a few markers was elevated (e.g. *Gfap* and *Cd68*) but most of them were unchanged (**new Supplementary Table S5 with specific References**). Moreover, we decided to check several toxicity markers in brain of FXTAS mouse which received ASO-CCG or saline. Western blot analysis did not show any changes in the level of GFAP, a marker of gliosis (both total and proteolytic fragment considered as the sign of glial injury¹), TNF- α , a marker of inflammation, CD68 and AIF1, markers of microglia and PPP1R1B, a marker of medium spiny neurons (**new Figure 5d**). However, we observed a trend towards an increase in the levels of GFAP proteolytic fragments in ASO-CCG treated mice. Therefore, we conclude that treatment with ASO-CCG indeed induces immune response in brain but induction is not severe, especially on a protein level.

In the new version of the manuscript we added new part to **Figure 1** concerning ASO-CCG effect on cells viability (**Figure 1c**) and apoptosis (**Figure 1d**). We also added new **Figure 5** and **Supplementary Table S5** where we present the influence of ASO-CCG on toxicity markers in mouse brain. Results concerning ASO-CCG toxicity in mice are described in new paragraphs of last subchapter.

Figure 1. Percent of PI positive (dead) COS-7 cells in variants treated with transfection agent (mock) or with different concentration of ASO-ctrl and ASO-CCG. None of comparisons are significant according to Student's t test.

• **Title includes “reversal”, which is a strong statement regarding the effect of the ASOs. This reviewer does not see evidence of reversal. Please provide this evidence. Overall, effects appear to be modest slowing of progression.**

We fully agree with the reviewer. We decided to change the title to “Short antisense oligonucleotides alleviate the pleiotropic toxicity of RNA harboring expanded CGG repeats”

• **Long-term effects on the mouse model have not been characterized**

o The authors note that “an interesting question to be addressed is whether this intervention is sufficient to provide long-term protection or merely slows down the pathogenic process.” The work would be improved by examining the long-term consequences of ASO treatment. **The authors should comment on feasibility of this test in their system and why not completed here.**

In our animal experiments we took advantage of a previously described inducible FXTAS P90CGG mouse model^{2,3}. In this model, a time course for the induction of the hallmark intranuclear inclusions and the development of a motor phenotype has been carefully established: It was shown that 12 weeks of doxycycline (DOX) induction, i.e. CGG^{exp} expression resulted in profound accumulation of aggregates in the cerebellum and FXTAS-like motor impairments. The phenotype is stably maintained even after cessation of the transgene expression, for at least 12 weeks³, which makes this time window ideally suited to investigate the effectiveness of intervention. Tissue collection in our study occurred more than two weeks after the end of DOX induction, i.e. more than 6 weeks in total after the end of ASO-CCG infusion (modified **Figure 4b**). It is considered that young adult mice age about 45 times faster than corresponding humans⁴, indicating that each week of our experiments reflects almost one year in men. While our results suggest that ASO-CCG treatment manifested in behavioral improvement, RNA-seq experiments and monitoring of number and size of FMRpolyG inclusions provided insights into effectiveness of this treatment in a relevant time frame. We have not systematically investigated the progression of phenotypes in the FXTAS P90CGG mouse model so far, but we typically observe that animals under transgene induction survive for more than one year. Based on the strongly encouraging findings of our current study we therefore believe that systematic preclinical assessment can be performed in this mouse model that takes into consideration the more long-lasting effects of ASO treatment and the influence of age on treatment efficacy. We point this out in the new version of Discussion on page 18, but we also believe that answering these questions experimentally is beyond the scope of the current study.

• **Mouse model improvements are not given in context of control mouse performance**

o **Rotarod performance data from control mice** in the authors' colony are not provided in Fig. 4c through Fig. 4e, so it is difficult to assess the severity of disease and magnitude of improvement in the treated and untreated model mice. **It would be helpful to describe the severity of disease in their model mice, including whether they die from their disease and the time course of their symptoms.**

Thank you very much for pointing this out. In this experiment we focused on FXTAS mouse with the hypothesis that a reduction of CGG^{exp} induced toxicity by ASO-CCG would be associated with an improvement of motor functions *in vivo* (page 10). Rotarod test has been employed as a standard test to evaluate motor performance in various FXTAS mouse models⁵⁻⁸. We have also previously

taken advantage of this test³ when describing the P90CGG mouse model and demonstrated a significantly reduced performance under comparable test conditions to the current study. In fact, DOX- control mice performed close to no-fault level in these tasks, whereas the P90CGG mice under 12 weeks DOX induction displayed significantly lower performance. Moreover, a shorter 8 weeks of DOX exposure did not produce these deficits. We explain this out now on page 11 and indicate the maximum cut-off levels of each of the rotarod tests in **Figures 4c, d and e** to illustrate the severity of the impairment in our FXTAS model and the relative improvement by the ASO treatment.

As mentioned above, we have previously further described the time course of phenotype development under these conditions and the stability of the pathology and behavioral phenotype after 12 weeks of transgene induction^{2,3}. We point this out now more clearly on page 10/11. Finally, we add a statement to our Discussion section that we have so far not obtained any evidence for an increased mortality of mice with transgene induction (page 18).

- **Claims of nuclear retention are not clearly supported**

- o The authors cautiously claim that the nuclear mRNA changes in Fig. 3c might suggest increased nuclear retention, but given the small effect size in the nucleus and larger change in the cytoplasm and the cell overall, it seems more likely that the nuclear changes occurred as a product of overall transcriptional changes.

We agree with the reviewer that a small increase of *FMR1* mRNA level in the nucleus observed in cell carrying CGG^{exp} most likely results directly from an elevated level of transcription. To strengthen this claim we decided to conduct additional experiments. We performed cellular fractionation on two additional fibroblast cell lines, one with normal (20 repeats) and one with expanded CGGs (90 repeats) in *FMR1* gene. Obtained data confirmed previous results (**Figure 3d**). We agree that our previous statement “Together, these findings suggest that ASO-CCG affects *FMR1* transcription, especially when it contains CGG^{exp}, and perhaps also the nuclear retention and stability of *FMR1* mRNA, leading to an increase in levels primarily in the cytoplasm” was confusing. However, It was not our intention to conclude that “the nuclear mRNA changes in Fig. 3c might suggest increased nuclear retention”. We changed all fragments which could somehow suggest that data shown on Fig. 3c suggest increase of nuclear retention. Based on obtained data we cannot, however, rule out a contribution of parallel/coexisting mechanisms including nuclear retention. In fact, according to our previous work concerning application of ASO targeting CUG repeats in the therapy of myotonic dystrophy type I where they decreased nuclear retention of mutated mRNA⁹ we assumed that in FXTAS model ASO-CCG may act similarly. As significantly higher increase of mRNA level was observed in cytoplasm we believe that an increase of mRNA stability could be a major driver of observed phenomenon. Increased stability of mRNA caused by reduced translation rate was described in a few paper including¹⁰⁻¹² and was discussed during several presentation at “RNA Meeting 2020”. Summing up, we believe that an increased level of mRNA of *FMR1* caused by the treatment with ASO-CCG has many sources and probably most important nuclear mechanism is transcription, whereas major cytoplasmic effect involves stability of mRNA.

To clarify this issue we changed our final statement in the new version of the manuscript (page 9/10 and 16). Moreover, we modified **Figure 3d** and added results for a two fibroblast cell lines.

- **Claims of R-loop suppression as the mechanism for transcriptional changes are not clearly supported**

- o The authors suggest that the *FMR1* transcriptional changes they observe occur due to R-loop suppression, but they fail to show clear evidence to support this. They show that the ASO's effect on

transcription appears RNase-dependent and they label an R-loop band in Fig. 3b, but clearly demonstrating that R-loop suppression specifically is the mechanism at play would require additional data and a more thorough explanation.

We agree with the Reviewer that we did not emphasize enough the relationship between ASO and R-loops. To meet mentioned doubts we added data in the main **Figure 3c** and in the Supplementary Information which we believe clarify this issue.

Firstly, the **Supplementary Figure S3b** presents that R-loops are formed in *FMR1* region containing CGG repeats what can be visualized in two ways: the first one assumes the use of undigested (circular DNA; original protocol described in¹³ and the second one the use of digested plasmid (linear DNA; our new protocol) containing the exon 1 of *FMR1* as a template for *in vitro* transcription (IVT). It turned out that our new approach can be successfully used to quantify efficiency of transcription and analyze R-loop formation *in vitro* (**new Supplementary Figures S3b, right panel; S3c; Figure 3b and new Figure 3c**).

Secondly, the use of fluorescently labelled ASO-CCG-Cy3 allowed us to monitor different RNA and DNA sequences containing CGG repeats. When IVT is performed in the presence of ASO-CCG-Cy3 we observe the signal coming from the Cy3, thus we know where ASO binds (**new Supplementary Figure S3c and new Figure 3c**). In our results we see that ASO-CCG-Cy3 binds to three different molecules/complexes: transcript (rCGG₁₀₀), ssDNA region containing CGG repeats of the DNA template and the R-loop structure. Signal from rCGG₁₀₀ and R-loop is sensitive to RNase A (digest all free RNA species), signal from ssDNA and R-loop is sensitive to DNase treatment, while signal from R-loop is sensitive to RNase H (digest RNA/DNA hybrids). Moreover we did several control experiments showing specificity of ASO-CCG in regulation of *in vitro* transcription of CGG₁₀₀ using several templates other than those containing CGG repeats. Control IVTs were not affected by ASO-CCG or RNase H treatment. Taken together these new data demonstrate that ASO-CCG indeed binds to RNase H-sensitive R-loops and to sense strand of DNA template containing CGG repeats, what can reduce stability of formed structures and what is associated with increase of *FMR1* transcription rate (**new Figure 3c**). Data coming from *in vitro* and cell-based experiments (here we added data from two additional cell lines, one with short and one with expanded CGGs in *FMR1*) showed that transcription of *FMR1* or its fragment containing expansion is sensitive to RNase H which recognizes and cleaves RNA within RNA/DNA duplex and positively regulates transcription efficiency (illustrated on **Figure 1a**).

In the new version of the manuscript we extended **Figure 3** and added **Supplementary Figure S3b and c**. Additional comments on those results were enclosed in a corresponding sections, mainly on page 9.

Minor

o Please provide more details regarding “ASO-ctrl” and rationale behind it

We designed ASO-ctrl as a random sequence of the same length and chemical modification as ASO-CCG. Considering the nucleotide composition of our target sequence (100% of GC), it would be risky to use control ASO with scrambled sequence, as it could still bind to CGG repeats. Designed ASO-ctrl did not show overt toxicity and its effect on studied molecular processes mostly resembled the results obtained from mock experiments (treatment of cells with transfection reagent only). The use of ASO-ctrl and obtained results ensured us that observed effects upon ASO-CCG treatment were not simply induced by the chemistry of oligomers and the process of administration of LNA oligomers into cells but arose from specific binding of ASO-CCG to the target sequence.

In the new version of the manuscript clarification of ASO-ctrl rationale is enclosed (page 5).

o Make it clear which aso is used in which experiments

Information regarding length of ASOs was stated in Materials and Methods section and indicated in legends to all Figures.

o Fig 1b: n? error bar meaning?

Kd values obtained in EMSA are expressed as the mean \pm SD, for N=3 for each ASO-CCG concentration.

o R-loops label on fig 3b is not obvious; should be explained in legend or elsewhere if authors really want to call attention to it

It is explained now in the description of **Supplementary Figure S3b** and **c**. We decided to show position of R-loop only for new experiment shown on **Figure 3c** and we removed the label from Figure 3b.

♣ Supposed to note reduction in R-loop band when ASO given?

In the case of experiments with ASO-CCG-Cy3 (**new Figure 3c**), we observe the signal coming from the Cy3, with or without RNase H pressure. Hence, we analyzed the effect of R-loops, which are sensitive to RNase H, on the transcription efficiency. We observed the positive effect on the transcription after R-loops digestion. In this approach the free ASO-CCG-Cy3 is visible on the gel (Cy3 filter), however free ASO-CCG cannot be detected (EtBr staining) (see **Figure 2** below). The ASO-CCG binding influences the visibility of bound nucleic acids preventing the quantitative measurement of the effect of ASO-CCG on the R-loop amount (**new Supplementary Figure S3c**). Hence, we are not able to quantify R-loop band after ASO treatment. Moreover, the experiments shown in this manuscript and our new experiments with ASOs targeting GC-rich sequences at both sides of CGG repeats (content of independent story) suggest that targeting CGG^{exp} did not significantly reduce efficiency of R-loop formation but significantly affect its thermodynamic stability.

Figure 2. **Left gel** IVT in the presence of ASO-CCG-Cy3, the gel was scanned and free ASO-CCG-Cy3 is detected. **Right gel** IVT in the presence of ASO-CCG, the gel was stained with EtBr and free ASO-CCG is not detected, it does not respond to EtBr staining. Also complexes of ASOs with RNA and DNA are not prone to be stained by intercalation of EtBr in a duplex region. Response of short single-stranded oligos and some chemically modified oligos to EtBr staining is highly sequence and modification dependent (short LNA not detected for example in¹⁴)

o Fig 3 c-e: why no data on control mice?

In this experiment we tested the hypothesis that a reduction of CGG^{exp} induced toxicity by ASO-CCG would be associated with an improvement of motor functions in a FXTAS mouse model (page 10). Previously we employed the rotarod test under test conditions comparable to those used here describing that while DOX- controls performed close to maximum cut-off/no-fault points of the test in these tasks, the FXTAS mouse model displays significantly lower levels of performance. This is now better explained on page 11. In addition, we now illustrate the maximum cut-off levels in **Figure 4c, d** and **e** in order to visualize the severity of the impairment in the mutant and the relative improvement by the ASO treatment.

References

1. Yang, Z. & Wang, K. K. W. Glial fibrillary acidic protein: From intermediate filament assembly and gliosis to neurobiomarker. *Trends Neurosci.* **38**, 364–374 (2015).
2. Hukema, R. K. *et al.* Reversibility of neuropathology and motor deficits in an inducible mouse model for FXTAS. *Hum. Mol. Genet.* **24**, 4948–4957 (2015).
3. Castro, H. *et al.* Selective rescue of heightened anxiety but not gait ataxia in a premutation 90CGG mouse model of Fragile X-associated tremor/ataxia syndrome. *Hum. Mol. Genet.* **26**, 2133–2145 (2017).
4. Flurkey, K., Curren, J. & Harrison, D. Mouse models in aging research. *Fac. Res.* 2000 - 2009 (2007).
5. Van Dam, D. *et al.* Cognitive decline, neuromotor and behavioural disturbances in a mouse model for fragile-X-associated tremor/ataxia syndrome (FXTAS). *Behav. Brain Res.* **162**, 233–239 (2005).

6. Qin, M. *et al.* A mouse model of the fragile X premutation: Effects on behavior, dendrite morphology, and regional rates of cerebral protein synthesis. *Neurobiol. Dis.* **42**, 85–98 (2011).
7. Sellier, C. *et al.* Translation of Expanded CGG Repeats into FMRpolyG Is Pathogenic and May Contribute to Fragile X Tremor Ataxia Syndrome. *Neuron* **93**, 331–347 (2017).
8. Wenzel, H. J. *et al.* Astroglial-targeted expression of the fragile X CGG repeat premutation in mice yields RAN translation, motor deficits and possible evidence for cell-to-cell propagation of FXTAS pathology. *Acta Neuropathol. Commun.* **7**, 27 (2019).
9. Wheeler, T. M. *et al.* Reversal of RNA dominance by displacement of protein sequestered on triplet repeat RNA. *Science (80-.)*. **325**, 336–339 (2009).
10. Mauger, D. M. *et al.* mRNA structure regulates protein expression through changes in functional half-life. *Proc. Natl. Acad. Sci. U. S. A.* **116**, 24075–24083 (2019).
11. Wu, Q. *et al.* Translation affects mRNA stability in a codon-dependent manner in human cells. *Elife* **8**, (2019).
12. Jia, L. *et al.* Decoding mRNA translatability and stability from the 5' UTR. *Nat. Struct. Mol. Biol.* (2020). doi:10.1038/s41594-020-0465-x
13. Groh, M., Lufino, M. M. P., Wade-Martins, R. & Gromak, N. R-loops Associated with Triplet Repeat Expansions Promote Gene Silencing in Friedreich Ataxia and Fragile X Syndrome. *PLoS Genet.* **10**, (2014).
14. Piao, X., Wang, H., Binzel, D. W. & Guo, P. Assessment and comparison of thermal stability of phosphorothioate-DNA, DNA, RNA, 2'-F RNA, and LNA in the context of Phi29 pRNA 3WJ. *RNA* **24**, 67–76 (2018).

Reviewers' Comments:

Reviewer #1:

Remarks to the Author:

The authors have done a great job in responding to the critiques and I believe it is a better paper now and it should be published now.

Reviewer #2:

Remarks to the Author:

Overall, the authors have addressed comments well.

Below are a few remaining suggestions/concerns.

- Fig 1. : Note in figure that cells used were from COS7 system described in methods and cited paper

- Fig 4 : Note in figure that cells used were from COS7 system described in methods and cited paper

- Fig 5 Western blot may be too imprecise to detect changes in immune markers in question. Consider qPCR on tissues from the same animals.

- Discussion: It doesn't seem correct to say that enhanced nucleocytoplasmic transport definitely isn't playing a role just because nuclear levels of mRNA don't change. Enhanced RNA production coupled with greater transport of transcripts from the nucleus to cytoplasm could produce the pattern observed here, increasing cytoplasmic transcript measurements while leaving nuclear measurements unchanged.

We thank Reviewers for considering our manuscript again. We revised the manuscript according to reviewers' comments. We believe that the new version of the manuscript is further improved after performing the recommended experiments and corrections.

Answers for specific remarks are specified bellow and changes in main text are highlighted using track changes feature in Microsoft Word.

REVIEWERS' COMMENTS

Reviewer #1 (Remarks to the Author):

The authors have done a great job in responding to the critiques and I believe it is a better paper now and it should be published now.

Thank you for your favorable opinion.

Reviewer #2 (Remarks to the Author):

Overall, the authors have addressed comments well.

Below are a few remaining suggestions/concerns.

- Fig 1. : Note in figure that cells used were from COS7 system described in methods and cited paper
- Fig 4 : Note in figure that cells used were from COS7 system described in methods and cited paper
- Fig 5 Western blot may be too imprecise to detect changes in immune markers in question. Consider qPCR on tissues from the same animals.
- Discussion: It doesn't seem correct to say that enhanced nucleocytoplasmic transport definitely isn't playing a role just because nuclear levels of mRNA don't change. Enhanced RNA production coupled with greater transport of transcripts from the nucleus to cytoplasm could produce the pattern observed here, increasing cytoplasmic transcript measurements while leaving nuclear measurements unchanged.

Thank you for your favorable opinion.

- Fig 1. : Note in figure that cells used were from COS7 system described in methods and cited paper
Information regarding COS7 cells used in experiments presented in Figure 1 is now stated in Figure legend.
- Fig 4 : Note in figure that cells used were from COS7 system described in methods and cited paper
Information regarding COS7 cells used in experiment presented in Figure 4 is now stated in Figure legend.

- Fig 5 Western blot may be too imprecise to detect changes in immune markers in question. Consider qPCR on tissues from the same animals.

We agree with the reviewer. Therefore, we decided to perform qPCR analysis of 5 immune system-related toxicity markers (*Gfap*, *Cd68*, *Aif1*, *Tlr3* and *Tlr7*) in cortex of saline- and ASO-CCG-treated P90CGG mouse. Two of this markers (*Gfap* and *Cd68*) was significantly upregulated in ASO-CCG treated mouse in accordance with our RNA-seq data, while three other remained unchanged (Supplementary Table S5). qPCR performed on samples derived from *N*=4 animals (each group contained 2 samples included and 2 samples not included to RNA-seq analysis) confirmed RNA-seq results. This new data are consistent with our previous statements that ASO-CCG induces an immune response in brain cortex, but this effect is not prominent. However, it is now explained more clearly in Results section (page 14) and in Discussion section (page 19) and results are presented as a new panel of Figure 5 (Fig. 5d).

- Discussion: It doesn't seem correct to say that enhanced nucleocytoplasmic transport definitely isn't playing a role just because nuclear levels of mRNA don't change. Enhanced RNA production coupled with greater transport of transcripts from the nucleus to cytoplasm could produce the pattern observed here, increasing cytoplasmic transcript measurements while leaving nuclear measurements unchanged.

We agree with the reviewer. Thus, in the revised version of the manuscript we changed previous statement (underlined sentence) from:

“One potential mechanism could be increased efficiency of nucleocytoplasmic transport, what we previously showed for mRNA containing CUG repeat expansion (rCUGexp) after treating muscles of myotonic dystrophy mice with short LNA-composed ASOs targeting CUG repeats. However, relatively low changes in mRNA level in nuclei of cells treated with ASO-CCG suggest that nucleocytoplasmic transport is not a major driver of significant differences in mRNA level in cytoplasm. According to recent findings, slowing down the ribosome movements or lowering the density of ribosomes on *FMR1* mRNA caused by ASO-CCG binding (Fig. 2e) may increase its half-life, what previously was shown for hundreds of mRNAs.”,

to:

“One potential mechanism could be increased efficiency of nucleocytoplasmic transport, what we previously showed for mRNA containing CUG repeat expansion (rCUG^{exp}) after treating muscles of myotonic dystrophy mice with short LNA-composed ASOs targeting CUG repeats. Other mechanisms could be linked to increased stability of mRNA. According to recent findings, slowing down the ribosome movements or lowering the density of ribosomes on *FMR1* mRNA caused by ASO-CCG binding (Fig. 2e) may increase its half-life, what previously was shown for hundreds of mRNAs.”